# Water-soluble 4-(dimethylaminomethyl) heliomycin exerts greater antitumor effects than parental heliomycin by targeting the tNOX-SIRT1 axis and apoptosis in oral cancer cells

Atikul Islam[1], Yu-Chun Chang[1], Xiao-Chi Chen[1], Chia-Wei Weng[1,2], Chien-Yu Chen[1], Che-Wei Wang[3,4], Mu-Kuan Chen[3,4], Alexander S Tikhomirov[5], Andrey E Shchekotikhin[5], Pin Ju Chueh[1,4,6,7]*

[1]Institute of Biomedical Sciences, National Chung Hsing University, Taichung, Taiwan; [2]Institute of Medicine, Chung Shan Medical University, Taichung, Taiwan; [3]Department of Otorhinolaryngology-Head and Neck Surgery, Changhua Christian Hospital, Changhua, Taiwan; [4]Department of Post-Baccalaureate Medicine, College of Medicine, National Chung Hsing University, Taichung, Taiwan; [5]Gause Institute of New Antibiotics, Moscow, Russian Federation; [6]Department of Medical Research, China Medical University Hospital, Taichung, Taiwan; [7]Graduate Institute of Basic Medicine, China Medical University, Taichung, Taiwan

*For correspondence:
pjchueh@dragon.nchu.edu.tw

Competing interest: The authors declare that no competing interests exist.

**Abstract** The antibiotic heliomycin (resistomycin), which is generated from *Streptomyces resistomycificus*, has multiple activities, including anticancer effects. Heliomycin was first described in the 1960s, but its clinical applications have been hindered by extremely low solubility. A series of 4-aminomethyl derivatives of heliomycin were synthesized to increase water solubility; studies showed that they had anti-proliferative effects, but the drug targets remained unknown. In this study, we conducted cellular thermal shift assays (CETSA) and molecular docking simulations to identify and validate that heliomycin and its water-soluble derivative, 4-(dimethylaminomethyl)heliomycin (designated compound 4-dmH) engaged and targeted with sirtuin-1 (SIRT1) in p53-functional SAS and p53-mutated HSC-3 oral cancer cells. We further addressed the cellular outcome of SIRT1 inhibition by these compounds and found that, in addition to SIRT1, the water-soluble 4-dmH preferentially targeted a tumor-associated NADH oxidase (tNOX, ENOX2). The direct binding of 4-dmH to tNOX decreased the oxidation of NADH to $NAD^+$ which diminished $NAD^+$-dependent SIRT1 deacetylase activity, ultimately inducing apoptosis and significant cytotoxicity in both cell types, as opposed to the parental heliomycin-induced autophagy. We also observed that tNOX and SIRT1 were both upregulated in tumor tissues of oral cancer patients compared to adjacent normal tissues, suggesting their clinical relevance. Finally, the better therapeutic efficacy of 4-dmH was confirmed in tumor-bearing mice, which showed greater tNOX and SIRT1 downregulation and tumor volume reduction when treated with 4-dmH compared to heliomycin. Taken together, our in vitro and in vivo findings suggest that the multifaceted properties of water-soluble 4-dmH enable it to offer superior antitumor value compared to parental heliomycin, and indicated that it functions through targeting the tNOX-$NAD^+$-SIRT1 axis to induce apoptosis in oral cancer cells.

## eLife assessment

This **useful** study reports that a water-soluble analog of heliomycin, 4-dmH, induces protein degradation of not only SirT1 but also tNOX, unlike heliomycin, which induces degradation of SirT1 but not tNOX, a difference that could in principle explain why 4-dmH induces apoptosis while heliomycin induces autophagy. The presented data provide **solid** support for the authors' conclusions.

## Introduction

Heliomycin (also known as resistomycin) was identified as a secondary metabolite produced by *Streptomyces resistomycificus* and has been reported to have antiviral (*Brazhnikova et al., 1958*; *Slesarchuk et al., 2020*), antifungal (*Zhang et al., 2013*), antibacterial (*Adinarayana et al., 2006*), antimicrobial, RNA synthesis-inhibiting (*Arora, 1985*), and HIV-1 protease-suppressing (*Roggo et al., 1994*) activities. It also possesses cytotoxicity against cancer cells from different tissues, such as cervical, gastric, hepatic, and breast cancers (*Adinarayana et al., 2006*; *Vijayabharathi et al., 2011*; *Riaz et al., 2020*; *Liu et al., 2018*). Interestingly, heliomycin isolated from marine sponges was reported to inhibit histone deacetylases (HDACs), as assessed by colorimetric assays and in silico docking studies (*Abdelfattah et al., 2018*). Among the HDACs, sirtuin-1 (SIRT1), a member in the Sir2 family of histone deacetylases, acts in different cellular compartments to remove acetyl groups from histone and non-histone proteins, and thereby modulate various cell functions, including metabolism, healthspan, tumorigenesis, apoptosis, and autophagy (*Xu et al., 2021*; *Garcia-Peterson and Li, 2021*; *Wang et al., 2021b*; *Vaquero et al., 2007*; *Houtkooper et al., 2012*; *Donmez and Guarente, 2010*; *Song and Surh, 2012*; *Choupani et al., 2018*; *Lee et al., 2008*). Given its position at the intersection between apoptosis and autophagy, it is not surprising that SIRT1 contributes to a complex regulatory network in which conventional signaling molecules undergo multifaceted interactions (*Sun et al., 2020*; *Karbasforooshan et al., 2018*; *Yousafzai et al., 2021*); however, no study has investigated the link between SIRT1 and heliomycin.

In an effort to identify cellular targets of heliomycin, we previously conducted a cellular thermal shift assays (CETSA), which is based on the concept that ligand binding augments the heat resistance of a target protein in intact cells (*Martinez Molina et al., 2013*; *Martinez Molina and Nordlund, 2016*). Data from CETSA and molecular docking simulations provided direct and experimental evidence linking heliomycin to SIRT1, which exhibited an enhanced melting temperature when incubated with heliomycin (*Lin et al., 2022*). Our results also demonstrated that heliomycin treatment downregulated SIRT1 expression and that the binding of heliomycin to SIRT1 triggered autophagy to repress the growth of bladder cancer cells. Given that the medical applications of heliomycin have been limited due to its extremely low solubility in aqueous media, we synthesized a series of chemically modified heliomycin derivatives. We found that their cytotoxicity was similar to that of the reference anticancer drug doxorubicin in several cancer cell lines (*Nadysev et al., 2018*; *Tikhomirov et al., 2021*; *Figure 1*). To continue with this line of investigation, we herein sought to elucidate the

**Figure 1.** Structures of heliomycin and 4-(dimethylaminomethyl)heliomycin (designated 4-dmH).

molecular mechanism and identify the cellular targets of the novel water-soluble 4-((dimethylamino) methyl)heliomycin (designated 4-dmH). We report that in addition to targeting SIRT1, as does its parental heliomycin, 4-dmH engages with the tumor-associated NADH oxidase (tNOX, ENOX2), as analyzed by CETSAs and molecular docking studies. Upon this binding, tNOX is ubiquitinylated and degraded, which in turn attenuates signaling by the tNOX-NAD$^+$-SIRT1 regulatory axis and induces apoptosis in oral cancer cells.

## Results

### Both heliomycin and its water-soluble derivative, 4-dmH, target intracellular SIRT1

We previously synthesized a series of heliomycin-derived Mannich bases with enhanced water solubility and demonstrated that they possessed anticancer activity against different cancer cell lines (*Nadysev et al., 2018*); however, the cellular targets and molecular mechanisms of heliomycin and these derivatives remained unknown. Very recently, SIRT1 was shown to function as a cellular target for heliomycin, and this targeting was identified as being important for the anticancer activity of heliomycin in bladder cancer cells (*Lin et al., 2022*). To assess whether the same was true for one of our novel water-soluble derivatives, 4-dmH, we first generated CETSA-based isothermal dose-response fingerprint curves (ITDRF$_{CETSA}$). Our results indicated that the heat resistance of SIRT1 was positively and dose-dependently associated with the water-soluble 4-dmH, with an OC$_{50}$ value of 1.0 µM for SAS cells and 0.9 µM for HSC-3 cells (*Figure 2a*). We next investigated the SIRT1-targeting ability of 4-dmH by determining the melting temperature ($T_M$) from CETSA melting curves, and found that the $T_M$ values were elevated from 47.0 °C (control) to 56.6 °C (4-dmH-exposed) in SAS cells, and from 47.2 °C (control) to 58.8 °C (4-dmH-treated) in HSC-3 cells (*Figure 2b*).

ITDRF$_{CETSA}$ also indicated that the parental heliomycin engaged with SIRT1 in oral cancer cells, as supported by the positive correlation of SIRT1 heat resistance with the heliomycin concentration (*Figure 3a*). Similarly, heliomycin treatment was associated with a greater than a 5 degree increase in $T_M$ values calculated from the CETSA melting curves of two oral cancer cell lines (*Figure 3b*). Our CETSA-based results suggested that the test compounds altered the unfolding and aggregation properties of SIRT1 in response to heat challenge, indicating that both 4-dmH and heliomycin biophysically interact with SIRT1 in oral cancer cells.

We next determined whether these compounds affect the in vitro deacetylase activity of SIRT1 using a fluorogenic substrate and recombinant protein, and found that both heliomycin and 4-dmH effectively inhibited SIRT1 deacetylase activity in vitro at 20 and 200 µM (*Figure 4a*). The degree of inhibition was comparable to that reported for the specific human SIRT1 inhibitor, sirtinol (IC$_{50}$, 131 µM) (*Mai et al., 2005*). Given that heliomycin was previously reported to target and downregulate SIRT1 in bladder cancer cells (*Lin et al., 2022*), in this present study, we addressed the question of SIRT1 inhibition by 4-dmH and its associated molecular events. We found that 4-dmH markedly attenuated SIRT1 protein expression at 1 and 2 µM for SAS cells and at 2 µM for HSC-3 cells (*Figure 4b*), and that pretreatment with the proteasome inhibitor, MG132, markedly reversed 4-dmH-inhibited SIRT1 expression in both cell lines (*Figure 4c*). Our immunoprecipitation results showed that 4-dmH increased the ubiquitination level of SIRT1 (*Figure 4d*), suggesting that 4-dmH may downregulate SIRT1 via ubiquitin-proteasomal degradation.

### Heliomycin provokes autophagy whereas 4-dmH triggers apoptosis

To further address the cellular outcome of 4-dmH-inhibited SIRT1, we explored the cell death pathway and signaling mechanisms involved in the system. As heliomycin was reported to induce autophagy in bladder cancer cells (*Lin et al., 2022*), we used AO staining (which detects acidic vesicular organelles) to assess whether heliomycin and/or water-soluble 4-dmH induced autophagy in oral cancer cells. Our results showed that heliomycin markedly enhanced autophagy in SAS cells at 1 and 2 µM and in HSC-3 cells at 2 µM; on the contrary, water-soluble 4-dmH largely failed to provoke autophagy in either line at concentrations up to 2 µM (*Figure 5a*). The lack of autophagic activity of 4-dmH prompted us to explore apoptosis, which is a cell death-related pathway previously reported to be triggered by a different heliomycin derivative in bladder cancer cells (*Lin et al., 2022*; *Nadysev et al., 2018*). Interestingly, we found that 2 µM of 4-dmH, but not the parental heliomycin, effectively

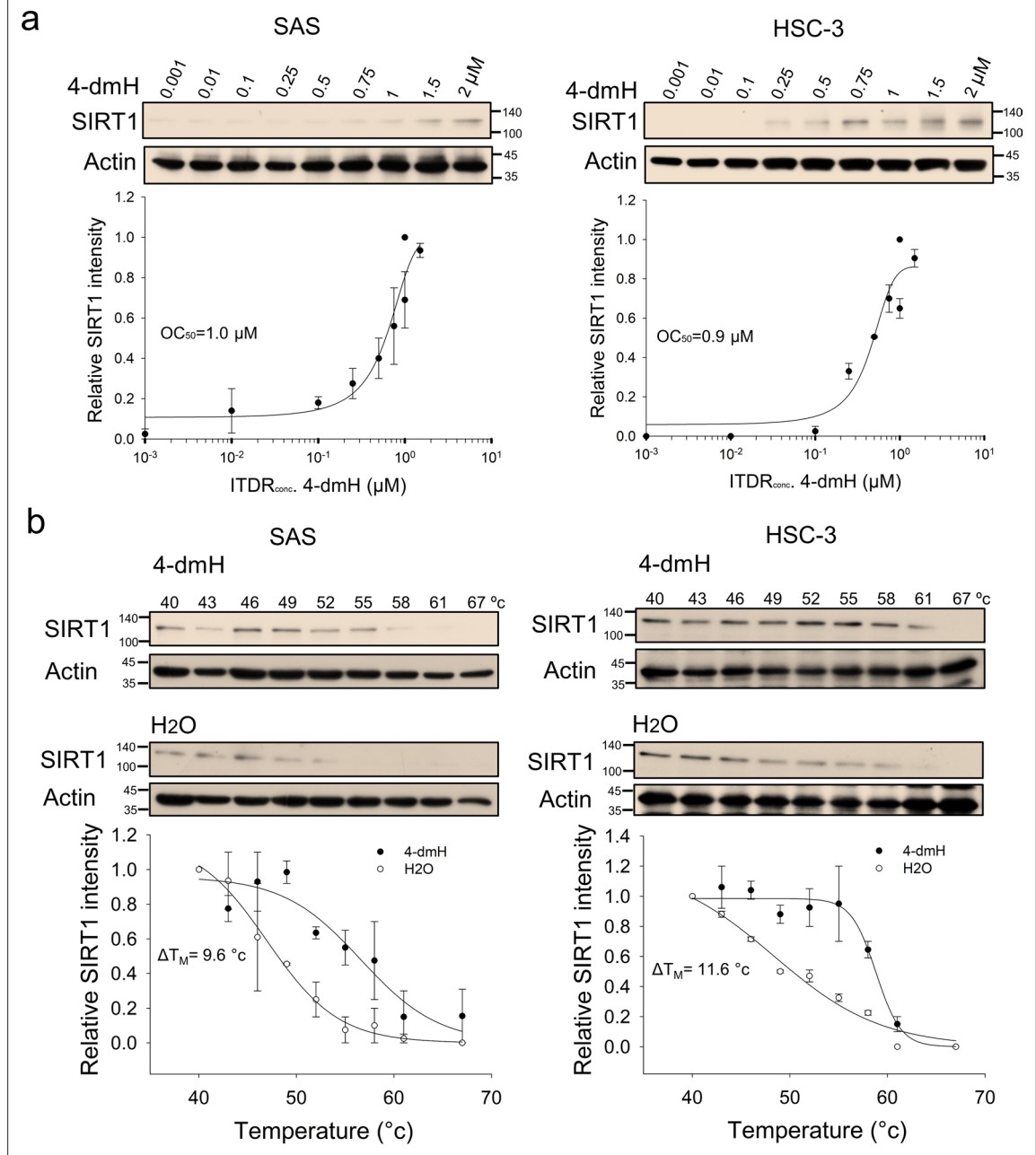

**Figure 2.** CETSA-based determination of binding between 4-dmH and SIRT1 protein. (**a**) Cells were incubated with different concentrations of 4-dmH as described in the Methods. Dose-dependent thermal stabilization of SIRT1 was assessed after heating samples at 54 °C for 3 min in SAS cells and HSC-3 cells. The band intensities of SIRT1 were normalized with respect to the intensity of actin. Representative chemiluminescence data are shown on the top. Data are presented as the average and values (mean ±SE) performed in triplicate (n=3). The representative images are generated by SigmaPlot followed by regression wizard. (**b**) CETSA-melting curves of SIRT1 in the presence and absence of 4-dmH in SAS cells and HSC-3 cells as described in the Methods. The immunoblot intensity was normalized to the intensity of the 40 °C samples. Representative chemiluminescence data are shown on the top. Data are presented as the average and values (mean ±SE) performed in triplicate (n=3). The representative images are generated by SigmaPlot followed by regression wizard. The denaturation midpoints were determined using a standard process.

The online version of this article includes the following source data for figure 2:

**Source data 1.** Original files for the western blot analysis in *Figure 2*.

**Source data 2.** Tiff files containing *Figure 2* of the relevant western blot analysis with the uncropped gels or blots with the relevant bands.

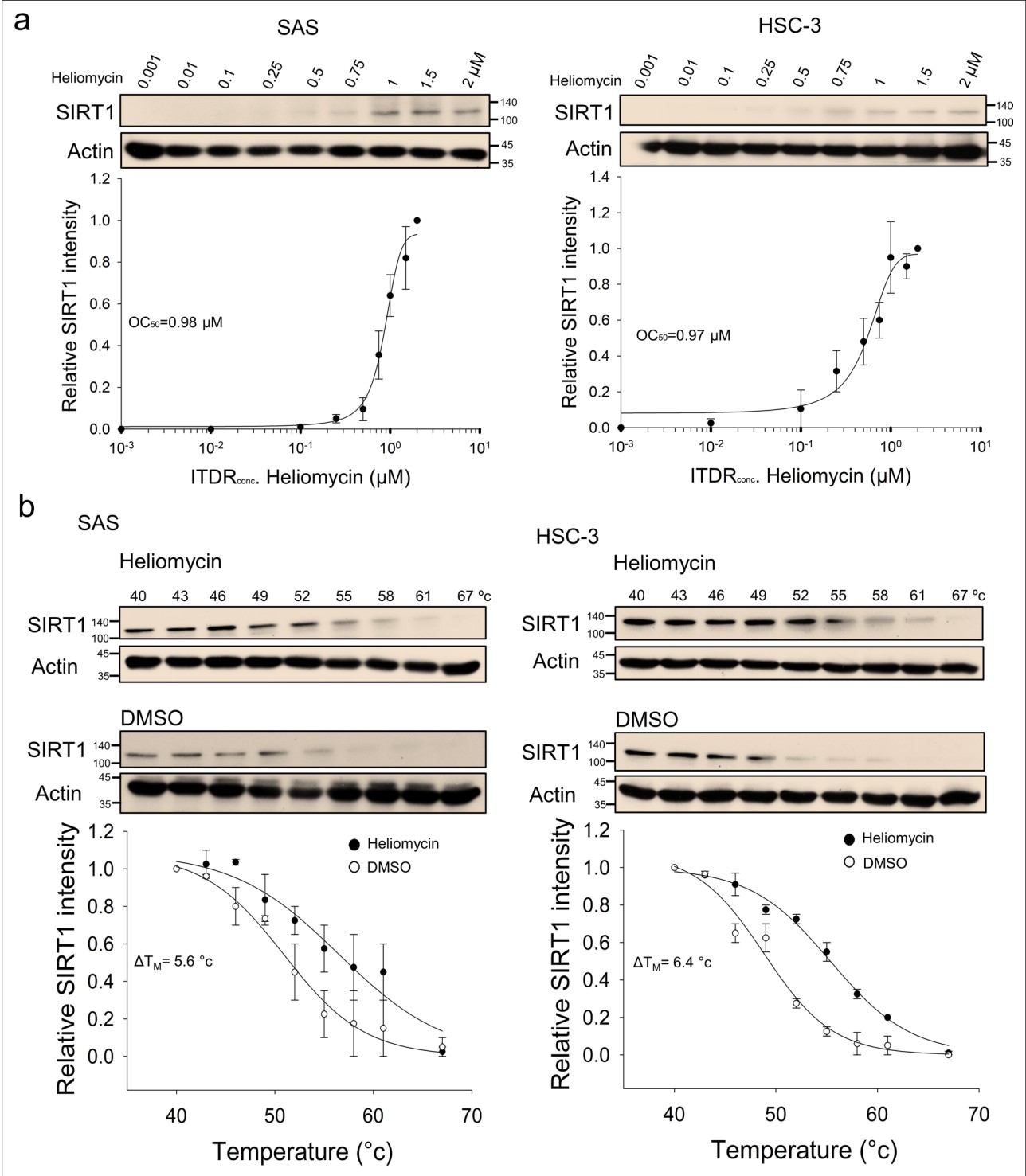

**Figure 3.** CETSA-based determination of binding between heliomycin and SIRT1 protein. (**a**) Cells were incubated with different concentrations of heliomycin as described in the Methods. Dose-dependent thermal stabilization of SIRT1 was assessed after heating samples at 54 °C for 3 min in SAS cells and HSC-3 cells. The band intensities of SIRT1 were normalized with respect to the intensity of actin. Representative chemiluminescence data are shown on the top. Data are presented as the average and values (mean ±SE) performed in triplicate (n=3). The representative images are generated by SigmaPlot followed by regression wizard. (**b**) CETSA-melting curves of SIRT1 in the presence and absence of heliomycin in SAS cells and HSC-3 cells as described in the Methods. The immunoblot intensity was normalized to the intensity of the 40 °C samples. Representative chemiluminescence data are shown on the top. Data are presented as the average and values (mean ±SE) performed in triplicate (n=3). The representative images are generated by SigmaPlot followed by regression wizard. The denaturation midpoints were determined using a standard process.

*Figure 3 continued on next page*

*Figure 3 continued*

The online version of this article includes the following source data for figure 3:

**Source data 1.** Original files for the western blot analysis in *Figure 3*.

**Source data 2.** Tiff files containing *Figure 3* of the relevant western blot analysis with the uncropped gels or blots with the relevant bands.

triggered JC-1 staining; this denotes a loss of mitochondrial membrane potential (MMP), which is an indicator of apoptosis (*Figure 5b*). The heliomycin-induced autophagy was accompanied by upregulation of autophagy markers, ULK1, Atg5/Atg7, and cleaved LC3-II, but downregulation of Bcl-2 (*Figure 5c* top). The ineffective apoptosis by heliomycin was evidenced by the downregulation of pro-apoptotic Bak and Puma and a lack of caspases 3-directed PARP cleavage and c-Flip downregulation (*Figure 5c* bottom). Alternatively, the water-soluble 4-dmH upregulated Bax, Noxa, Puma, and PARP cleavage; this suggested the possible relevance of a p53-dependent apoptotic pathway in these cells (*Figure 5d*). Noteworthily, the 4-dmH-induced downregulation of SIRT1 was concurrent with an increase in c-Myc acetylation, leading to downregulation of anti-apoptotic c-Flip in p53-mutated HSC-3 cells, suggesting the possible involvement of a p53-independent pathway (*Figure 5d* right). The absence of upregulation of ULK1, Atg 5, Atg7, and cleaved LC3-II provided evidence for the inadequate autophagy by 4-dmH (*Figure 5d* bottom).

Given that autophagy can either support survival or cause cell death, we analyzed the cellular outcome of heliomycin-induced autophagy by cell impedance measurements. Our results demonstrated that heliomycin decreased cell proliferation in both oral cancer cell lines, although the induced cytotoxic effect was less evident at 1 µM in HSC-3 cells. However, heliomycin did not exhibit cytotoxicity toward non-cancerous human BEAS-2B cells (*Figure 6a*). The water-soluble 4-dmH also effectively diminished cell proliferation in a dose-dependent manner in oral cancer cells, but not in that of BEAS-2B cells (*Figure 6b*). The preferential cytotoxicity of 4-dmH toward cancer cells was also reported in our previous study, indicating that 4-dmH displayed much higher $IC_{50}$ values against non-cancerous human dermal microvascular endothelium HMEC-1 cells compared to those of tumor cells (*Nadysev et al., 2018*). Additionally, our results from colony-forming assays revealed that both compounds exhibited high growth-suppressive ability against oral cancer cells (*Figure 6c*). Nevertheless, the lack of changes in cell phase population indicated that the diminution in cell growth by heliomycin and 4-dmH was least likely to arise from cell cycle arrest (*Figure 6d*).

## 4-dmH, but not heliomycin, targets intracellular tNOX, an upstream regulator of SIRT1

We next addressed the molecular mechanisms underlying SIRT1 inhibition and concurrent cell death by these two compounds in oral cancer cells. Being an $NAD^+$-dependent protein deacetylase, SIRT1 activity is primarily governed by $NAD^+$/NADH ratio, thus, there exists a positive correlation between these two (*Mouchiroud et al., 2013*; *He et al., 2022*; *Donmez, 2012*; *Teertam and Phanithi, 2022*; *Li et al., 2023*; *Bai et al., 2011*; *Ma et al., 2015*; *Fulco et al., 2003*; *Yang et al., 2022*). We questioned whether these two compounds inhibit SIRT1 by affecting the intracellular $NAD^+$/NADH levels, and were surprised to find that 4-dmH, but not heliomycin, caused a prominent inhibition of intracellular $NAD^+$/NADH ratio (*Figure 7a*). The discrepancy in their ability to reduce $NAD^+$ generation led us to explore the role of a tumor-associated NADH oxidase (tNOX, ENOX2) in 4-dmH-suppressed SIRT1 and apoptosis induction. We have previously reported that tNOX inhibition reduced the intracellular $NAD^+$/NADH ratio and SIRT1 deacetylase activity, increasing p53 acetylation and apoptosis (*Tikhomirov et al., 2015*; *Chang et al., 2020*; *Lin et al., 2019*; *Chen et al., 2017*). In the light of this information, we assessed the effect of the compounds on tNOX expression and found that 4-dmH, but not heliomycin, considerably diminished tNOX protein expression in a concentration-dependent manner (*Figure 7b*). Pretreatment with the proteasome inhibitor, MG-132, competently recovered the tNOX downregulation triggered by 1 or 2 µM 4-dmH in both cell lines, suggesting that the proteasomal degradation pathway may be associated with 4-dmH-mediated tNOX downregulation (*Figure 7c*). Consistently, 4-dmH increased the ubiquitination level of tNOX, suggesting the possible involvement of ubiquitination- and proteasome-driven protein degradation in the 4-dmH-induced downregulation of tNOX (*Figure 7d*). The RNAi-mediated knockdown of tNOX significantly triggered

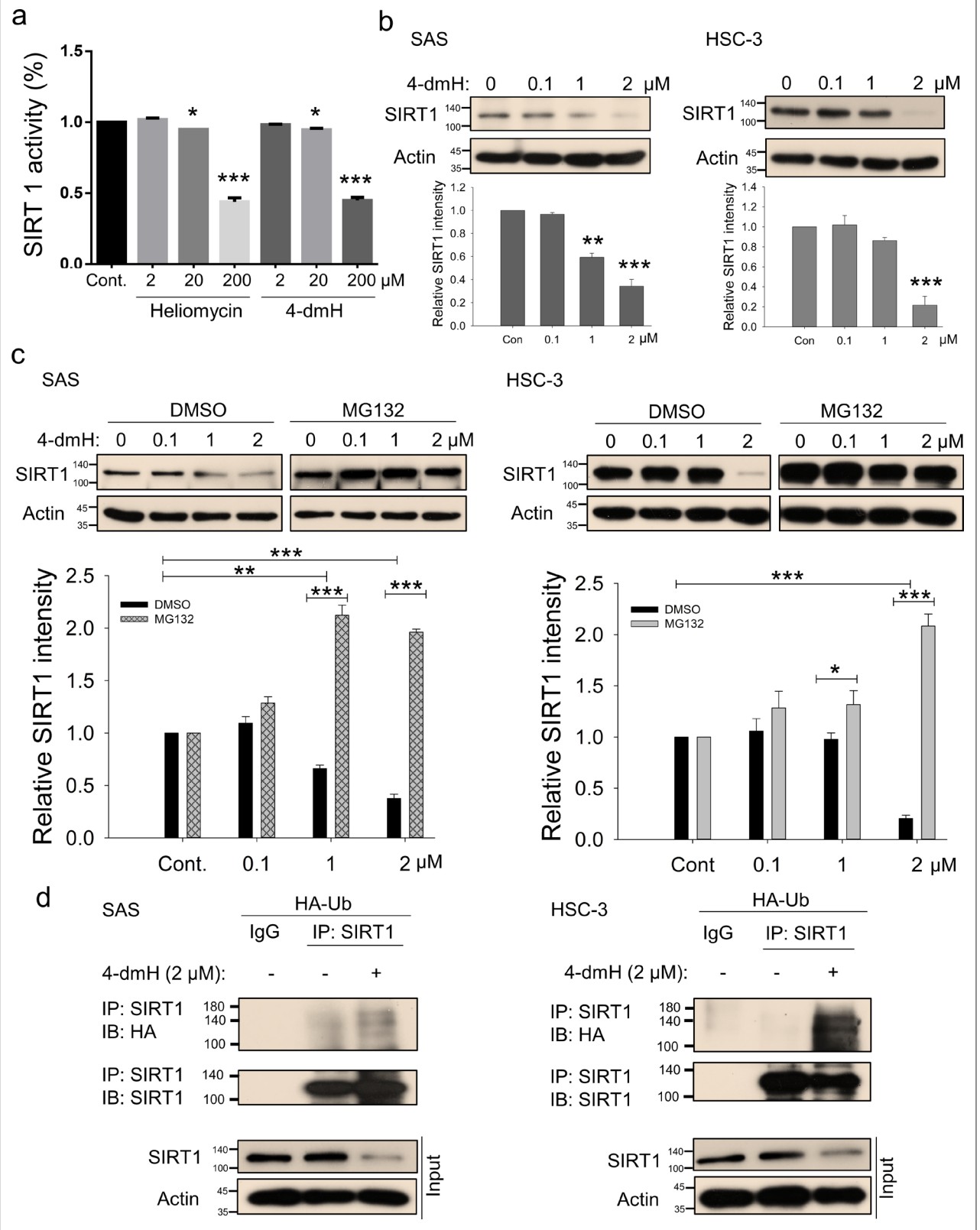

**Figure 4.** SIRT1 inhibition and downregulation by water-soluble 4-dmH. (**a**) SIRT1 activity was determined by a SIRT1 Activity Assay Kit (Fluorometric) with control or recombinant protein treated with test compounds. Values (mean ±SE) are from three independent experiments (n=3). The significance of differences between control and treatment groups was calculated using a one-way ANOVA followed by an appropriate post-hoc test such as LSD. The significant values are as *p<0.05, ***p<0.001. (**b**) 4-dmH markedly attenuated SIRT1 protein expression analyzed by western blotting. Values (mean

*Figure 4 continued on next page*

*Figure 4 continued*

±SE) are from three independent experiments (n=3). The significance of differences between control and treatment groups was calculated using a one-way ANOVA followed by an appropriate post-hoc test such as LSD. The significant values are as **p<0.01, ***p<0.001. (**c**) 4-dmH-suppressed SIRT1 expression was reverted by MG132, the proteasome inhibitor, in SAS cells and HSC-3 cells. Aliquots of cell lysates were resolved by SDS-PAGE and analyzed by western blotting. β-Actin was detected as an internal control. Representative images are shown. Values (means ± SDs) are from three independent experiments (n=3). The significance of differences between control and treatment groups was calculated using a one-way ANOVA followed by an appropriate post-hoc test such as LSD. The significant values are as *p<0.05, **p<0.01, *** p<0.001. (**d**) The lysates of HA-Ub overexpressing cells were immunoprecipitated with nonimmune IgG or an antibody against SIRT1, and the bound proteins were detected by western blotting with anti-HA or anti-SIRT1 antibodies. The total lysates were also immunoblotted with anti-SIRT1 or anti-β-actin antibodies. Aliquots of cell lysates were resolved by SDS-PAGE and analyzed by western blotting. β-actin was detected as an internal control. Representative images are shown.

The online version of this article includes the following source data for figure 4:

**Source data 1.** Original files for the western blot analysis in *Figure 4*.

**Source data 2.** PDF containing *Figure 4b-d* of the relevant western blot analysis with the uncropped gels or blots with the relevant bands.

spontaneous autophagy and apoptosis in SAS cells (*Figure 7e*), further validating its role in the regulation of cell death.

Using CETSA, we explored whether tNOX could be a cellular target of 4-dmH. Our results demonstrated that the heat stability of tNOX was found to be dose-dependently enhanced by 4-dmH (*Figure 8a*), indicating that there was direct binding between the protein and tested compound. The 4-dmH treatment increased the $T_M$ value from 45.7 °C (control) to 50.9 °C (4-dmH-incubated) in SAS cells, and from 47.7 °C (control) to 56.9 °C (4-dmH-incubated) in HSC-3 cells (*Figure 8b*), further corroborating this direct interaction. In contrast, 4-dmH treatments did not seem to increase the melting temperature of PARP or the nicotinamide adenine dinucleotide phosphate (NADPH) oxidase-4 (NOX4), excluding those two proteins as potential targets of 4-dmH (*Figure 8c*). Similarly, our results from CETSA indicated that the parental heliomycin did not directly bind to tNOX in either cell line (*Figure 8d*).

## Molecular docking simulation predicts better binding of heliomycin with SIRT1, and 4-dmH with tNOX protein

To study the binding mode of 4-dmH and its parental heliomycin in the binding pockets of the protein structures of SIRT1 and tNOX, we used molecular docking simulations. The binding poses of heliomycin and 4-dmH were predicted to have similar orientations for SIRT1 (*Figure 9a*). The docking energy scores indicated that the heliomycin-SIRT1 complex (–10.1 kcal/mol) exhibited higher affinity than the 4-dmH-SIRT1 complex (–8.0 kcal/mol). The heliomycin-SIRT1 and 4-dmH-SIRT1 complexes were mainly stabilized with hydrophobic interactions, with three consistent H-bonds predicted in the binding pocket (red circles and ellipses, *Figure 9b*). However, although the 3-hydroxyl, 7-hydroxyl, and 2-carbonyl groups of heliomycin and 4-dmH consistently formed hydrogen bonds with Ala262, Arg274, and Ile347, respectively, the 4-dmH-SIRT1 complex exhibited an additional H-bond with Ser441 in the side 4-dimethylaminomethyl group (*Figure 9b*). The additional H-bond was predicted to decrease ligand affinity to SIRT1's binding cavity due to its interference with the hydrophobic interactions.

Next, we performed a docking study of heliomycin and 4-dmH with tNOX. Previous reports showed that doxorubicin (adriamycin) could inhibit tNOX activity (*Morré et al., 1997*; *Hedges et al., 2003*). Therefore, we simulated the binding poses of heliomycin, 4-dmH, and doxorubicin (as positive control) using a blind docking approach. The results of our computational simulations showed that heliomycin, 4-dmH, and doxorubicin could bind in the same pocket of tNOX protein, but with different orientations of their core moieties (*Figure 9c*). The docking energy scores of the 4-dmH-tNOX and doxorubicin-tNOX complexes were similar to one another (–8.5 and –8.4 kcal/mol, respectively), while the heliomycin-tNOX complex had a slightly lower affinity (–8.1 kcal/mol). A number of consistent interaction residues were found in the three docked complexes, including Ile90, Lys98, Pro111, Pro113, Leu115, Pro117, and Pro118 (*Figure 9d*). Both heliomycin and 4-dmH produced three H-bonds, but the flipped orientation of 4-dmH yielded a higher number of hydrophobic interactions than the heliomycin-tNOX complex. This suggests that the hydrophobic moieties surrounding 4-dmH yield a higher affinity for tNOX protein.

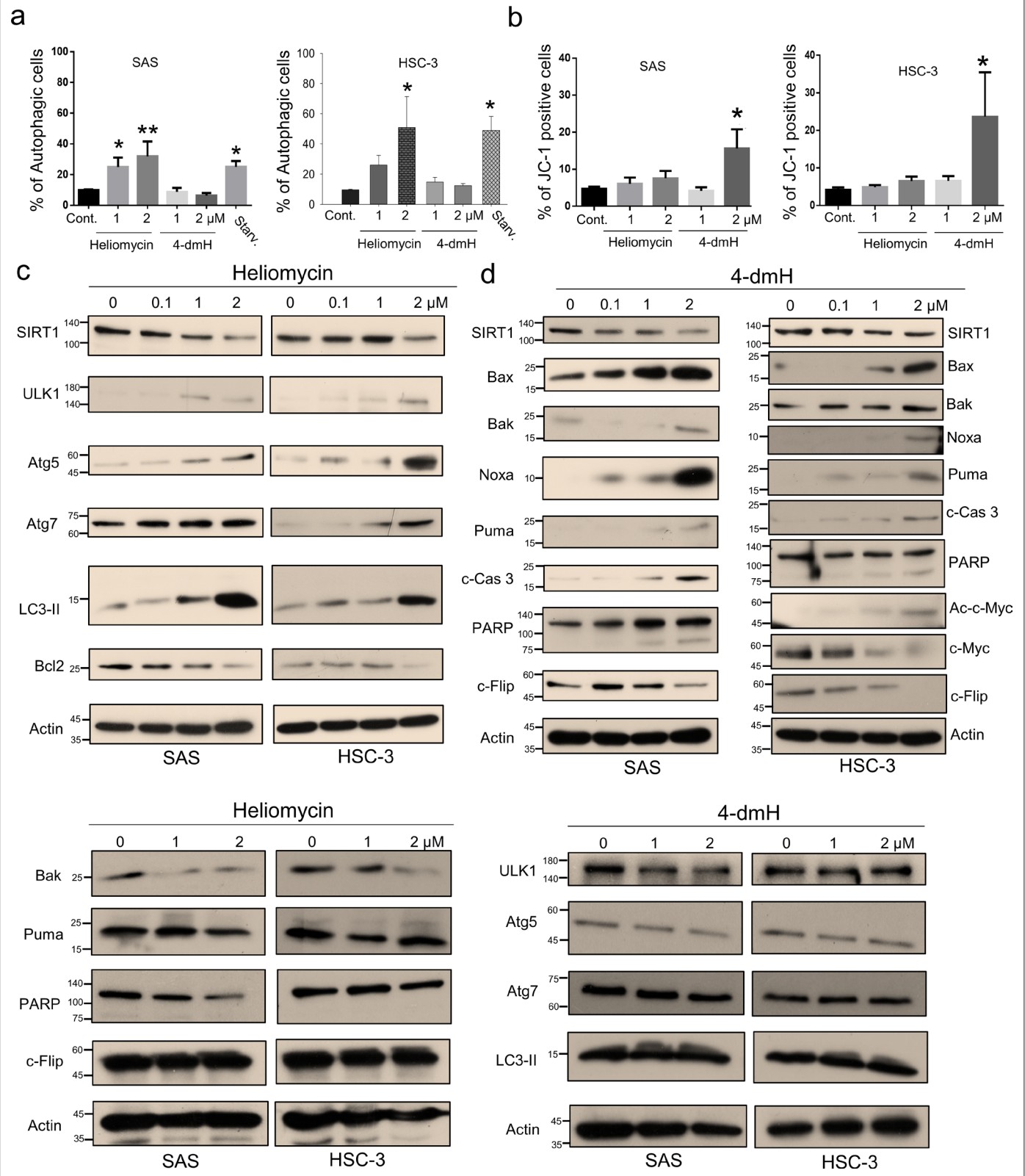

**Figure 5.** Heliomycin and 4-dmH provoked different cell death pathways in oral cancer cells. (**a**) Cells were exposed to heliomycin or 4-dmH and the percentage of the autophagic population was determined by AO staining using flow cytometry. The results are expressed as a percentage of autophagic cells. Values (mean ±SE) are from three independent experiments (n=3). The significance of differences between control and treatment groups was calculated using a one-way ANOVA followed by an appropriate post-hoc test such as LSD. The significant values are as *p<0.05, **p<0.01.

*Figure 5 continued on next page*

*Figure 5 continued*

(**b**) Cells were exposed to heliomycin or 4-dmH and the percentage of the apoptotic population was determined by JC-1 staining using flow cytometry. The results are expressed as a percentage of autophagic cells. Values (mean ±SE) are from three independent experiments (n=3). The significance of differences between control and treatment groups was calculated using a one-way ANOVA followed by an appropriate post-hoc test such as LSD. The significant value is as *p<0.05. (**c, d**) Cells were treated with heliomycin for various concentrations, and aliquots of cell lysates were resolved by SDS-PAGE and analyzed for protein expression by western blotting. β-actin was used as an internal loading control to monitor for equal loading.

The online version of this article includes the following source data for figure 5:

**Source data 1.** The flow cytometry dot blot autophagy data in *Figure 5a*.

**Source data 2.** The flow cytometry dot blot JC-1 staining data in *Figure 5b*.

**Source data 3.** Original files for the western blot analysis in *Figure 5*.

**Source data 4.** PDF containing *Figure 5c,d* of the relevant western blot analysis with the uncropped gels or blots with the relevant bands.

Furthermore, for further evaluation of the importance of the consistent interaction residues in the three docked compound-tNOX complexes, the seven interaction residues on tNOX were substituted with alanine or glycine amino acids and then simulated the protein structures. The simulated protein structures appear slightly different from the original tNOX structure. Overall, the root mean square difference between the original tNOX structure and the structures with residues substituted by alanine or glycine amino acids was estimated at 3.339 or 4.024 angstroms (Å), respectively (*Figure 9—figure supplement 1a*). The simulated protein structures were also employed to conduct the docking analysis for 4-dmH. The results of further docking analysis revealed that 4-dmH could bind within the same pocket of different types of tNOX structures but with varying orientations (*Figure 9—figure supplement 1b*). This observation also suggests that the replacement of both key residues with alanine or glycine could result in a reduction of the binding affinity of 4-dmH to tNOX, with values of –8.2 and –7.6 kcal/mol, respectively. Moreover, the substitution of both key residues with alanine or glycine also reduces the number of the original interacting residues and interaction forces in core moieties in the 4-dmH-tNOX complexes (*Figure 9—figure supplement 1c and d*). Together, our experimental results and molecular docking simulations are consistent with the notion that 4-dmH possesses a better affinity ability for tNOX than for SIRT1.

## 4-dmH exhibits greater therapeutic efficacy in vivo

Given our observations indicating that SIRT1 and tNOX are cellular target(s) for heliomycin or 4-dmH, we next investigated the therapeutic effects of these compounds in vivo. Mice were inoculated with SAS cells, and tumor-bearing animals (average tumor diameter 0.5–1 cm) were randomly split into three groups and intratumorally injected with buffer, 200 μg heliomycin in water, or 200 μg 4-dmH in vehicle buffer. We found that 4-dmH treatment yielded a significant reduction in tumor sizes at day 6, and continued to do so until animals were terminated at day 10 (*Figure 10a*). By comparison, heliomycin treatment appeared to be less effective in diminishing tumor growth compared to 4-dmH (*Figure 10a and b*). Protein analysis of mouse tumor tissues further showed that 4-dmH exhibited greater inhibition of tNOX expression compared to heliomycin, but the two compounds were similar in their abilities to downregulate SIRT1 expression (*Figure 10c*). Interestingly, tNOX knockdown was reported to partially attenuate SIRT1 expression and repressed growth in various cancer cell types, such as lung (*Lee et al., 2015*), bladder (*Lin et al., 2016*), and stomach (*Chen et al., 2017*). We also observed that tNOX is acetylated/ubiquitinated under certain stresses and the SIRT1 depletion affects tNOX expression (data not shown). It is speculated that SIRT1 deacetylates tNOX and modulates its protein stability. Thus, there is a reciprocal regulation between tNOX and SIRT1. Although heliomycin does not bind to tNOX, its inhibition of SIRT1 activity/expression might have an impact on tNOX expression. Consistent with our in vitro data, heliomycin enhanced the levels of autophagy-associated markers, such as Atg 5, Atg 7 and cleaved LC3. 4-dmH treatment was associated in vivo with upregulations of pro-apoptotic Bak and caspase 3-directed PARP cleavage, and the concurrent downregulation of anti-apoptotic Flip (*Figure 10c*), suggesting that the greater inhibition of tumor volume by this water-soluble derivative of heliomycin was related to apoptosis induction.

Finally, we explored the potential clinical relevance of tNOX and SIRT1 protein in oral cancer. Protein analysis of tissues from oral cancer patients indicated that both tNOX and SIRT1 were upregulated in four out of five tumor samples compared to their adjacent normal tissues (*Figure 10d*).

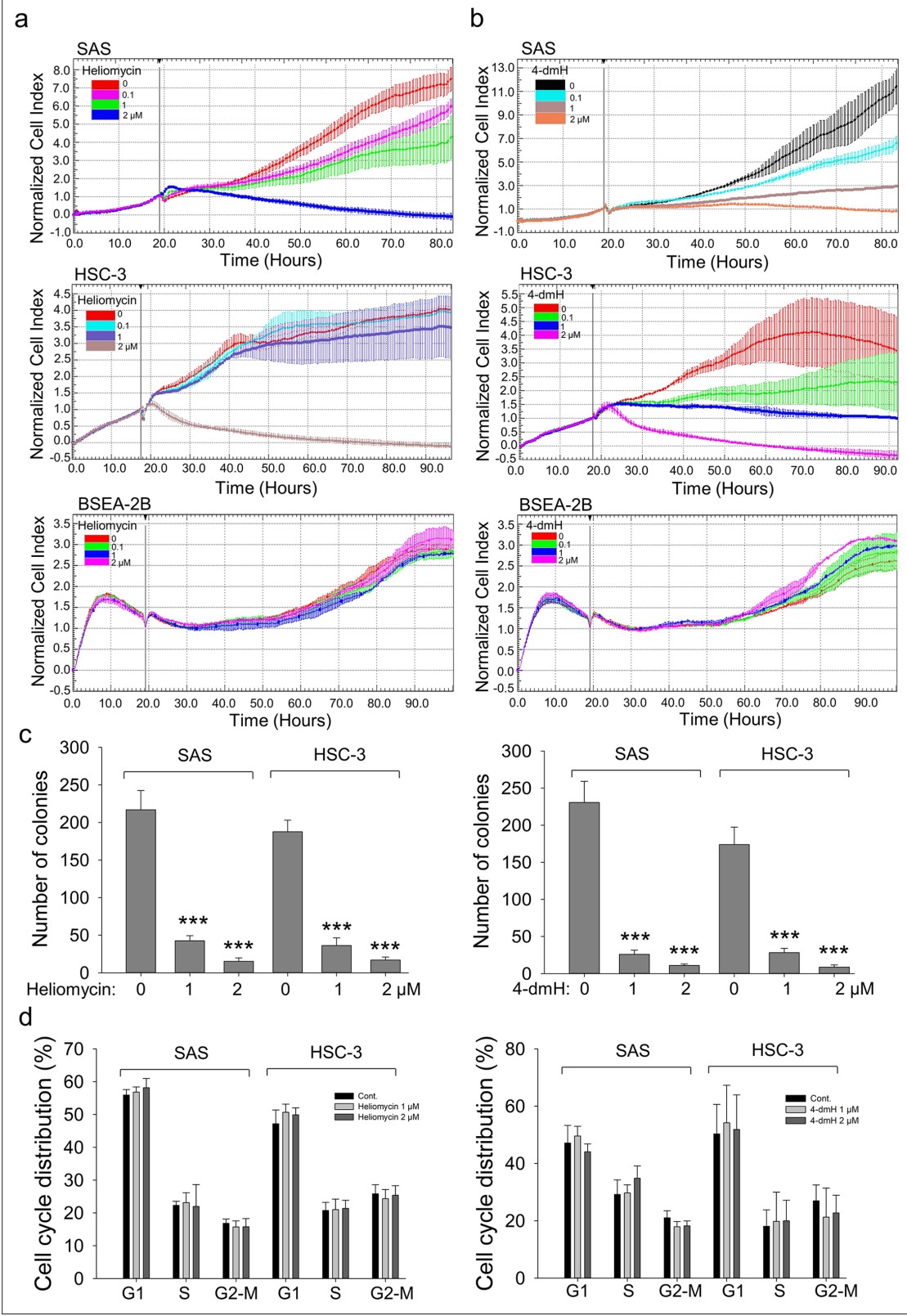

**Figure 6.** The effects of heliomycin and 4-dmH on proliferation, colony formation, and cell cycle progression of oral cancer cells. (**a, b**) Cell proliferation was dynamically monitored by impedance measurements in SAS, HSC-3 cells, and BSEA-2B cells as described in the Methods. Shown are the normalized cell index values measured. (**c**) Cells were treated with various concentrations of compounds and allowed to form colonies. Colony numbers were determined and documented. Values (means ± SDs) are from three independent experiments (n=3). There was a significant reduction in cells

*Figure 6 continued on next page*

*Figure 6 continued*

treated with heliomycin or 4-dmH compared to the controls. The significance of differences between control and treatment groups was calculated using a one-way ANOVA followed by an appropriate post-hoc test such as LSD. The significant value is as *** p<0.001. (**d**) Cells were exposed to different concentrations of compounds for 24 hr and assayed for the cell cycle phase using flow-cytometry. The graphs are representative of three independent experiments (n=3) and generated by SigmaPlot.

The online version of this article includes the following source data for figure 6:

**Source data 1.** The colony formation representative data in *Figure 6c*.

**Source data 2.** The flow cytometry dot blot cell cycle analysis data in *Figure 6d*.

Data mining in the Kaplan-Meier plotter (pan-cancer RNA-seq dataset; https://www.kmplot.com/), revealed that high tNOX expression was associated with a poor prognosis for overall survival [hazard ratio (HR): 2.81, log-rank p=0.008] among 78 stage-III head-neck cancer patients. The median overall survival in the low tNOX expression cohort was 213.9 months, compared to 28.7 months in the high tNOX expression cohort (*Figure 10e*). However, we did not find any significant correlation between tNOX and the survival rate in the earlier stages of the disease. The expression of SIRT1 did not appear to significantly contribute to predicting overall survival at any stage of head-neck cancer.

In sum, we herein show that in addition to targeting SIRT1, as does its parental heliomycin, the enhanced water solubility and preferential targeting of tNOX by 4-dmH grants it superior therapeutic value compared to heliomycin. Upon this binding, tNOX is ubiquitinylated and degraded, which in turn attenuates signaling by the tNOX-NAD$^+$-SIRT1 regulatory axis and induces apoptosis in oral cancer cells, as evidenced by the results of in vitro and in vivo studies.

## Discussion

Oral cancers, a disease of epithelial origin, are commonly found in the lining of the lips, mouth, and upper throat. The updated data in GLOBOCAN 2020 estimated that there were nearly 0.38 million new cases and over 0.17 million deaths worldwide from cancers in the lip and oral cavity (*Sung et al., 2021*). Unfortunately, the prognosis and 5-year overall survival of oral cancer patients remain disappointing, despite countless efforts to improve early diagnosis and expand treatment options (*Sieviläinen et al., 2019*; *Lacas et al., 2017*; *Carvalho et al., 2005*). In conjunction with surgery, radio-therapy, and chemotherapy, targeted therapy has emerged as a promising therapy for oral cancers, such as by targeting proteins that are overexpressed in cancer cells (e.g. EGFR and PD1; *Li et al., 2022*; *Cheng et al., 2022*; *Cao et al., 2022*). In this regard, we explored the possibility of the antibi-otic heliomycin and a water-soluble heliomycin derivative as therapeutic strategies and identified their cellular targets. Here, we report that the water-soluble heliomycin derivative, 4-dmH exhibits a higher affinity for tNOX compared to its parental heliomycin, and that tNOX has a stronger affinity than SIRT1 for the 4-dmH. In addition to the action of 4-dmH on SIRT1, its preferential targeting of tNOX resulted in greater therapeutic efficacy in diminishing oral cancer cell proliferation and inhibiting tumor growth by inhibiting tNOX activity, increasing its protein degradation, and ultimately provoking apoptosis in p53-functional and p53-mutated oral cancer systems (*Figure 10f*).

Heliomycin (also called resistomycin) was originally isolated from marine sponges and shown to possess a wide range of activities, including HDAC inhibitor activity (*Abdelfattah et al., 2018*). HDAC inhibitors are currently under investigation for playing important roles in cancer epigenetic pathways, exhibiting antitumor activity, and decreasing tumor resistance (*Basha and Basavarajaiah, 2022*; *Sun et al., 2022*; *Contreras-Sanzón et al., 2022*). Our group has explored the anticancer properties of heliomycin and established its binding with SIRT1 in the native cellular environment of bladder cancer cells (*Lin et al., 2022*; *Nadysev et al., 2018*). SIRT1, which belongs to class III of the HDAC super-family, has the unique feature of being NAD$^+$-dependent; it contributes to inflammatory oral diseases and oral cancer by modulating many transcription factors and antioxidant enzymes (*Pan et al., 2022*). In the present study, we observed upregulation of SIRT1 in tumor tissues of animals and oral cancer patients (*Figure 10*). However, data mining using a pan-cancer RNA-seq dataset (https://www.kmplot.com/) suggests that SIRT1 expression does not have significant importance in predicting overall survival at any stage of head-neck cancer. A potential correlation between SIRT1 expression and prognosis for head-neck cancer may warrant further discussion. Using molecular docking simu-lations and cell-based studies, we herein demonstrated that both 4-dmH and heliomycin targeted

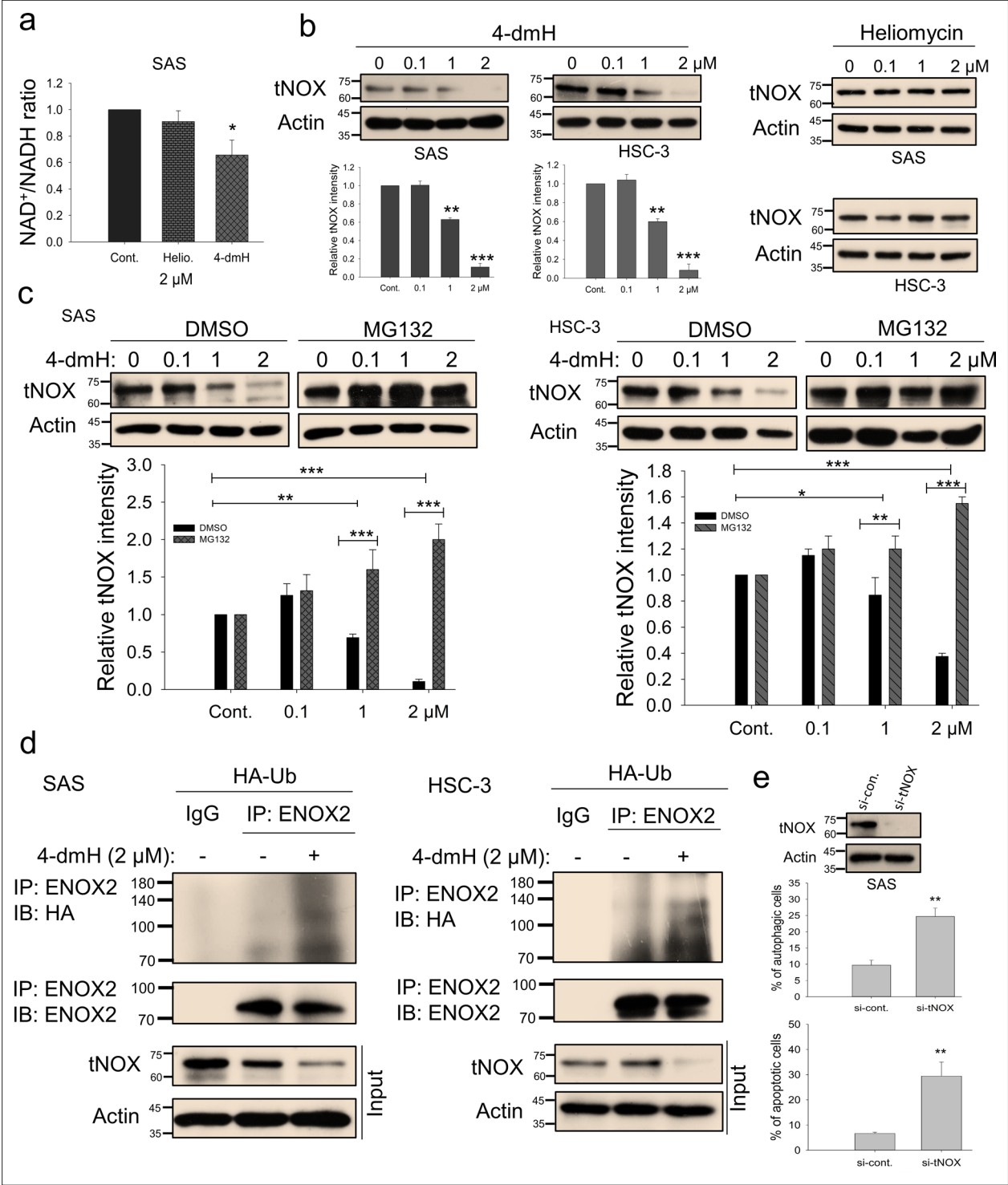

**Figure 7.** tNOX inhibition and downregulation by water-soluble 4-dmH. (**a**) The intracellular NAD$^+$/NADH ratio was measured by an NADH/NAD Quantification Kit (BioVision Inc) with control or lysates of SAS cells treated with heliomycin or 4-dmH. Values (mean ±SE) are from three independent experiments (n=3). The significance of differences between control and treatment groups was calculated using a one-way ANOVA followed by an appropriate post-hoc test such as LSD. The significant value is as *p<0.05. (**b**) 4-dmH, but not heliomycin, markedly attenuated tNOX protein expression analyzed by western blotting. Values (mean ±SE) are from three independent experiments (n=3) for 4-dmH groups. The significance of differences between control and treatment groups was calculated using a one-way ANOVA followed by an appropriate post-hoc test such as LSD. The significant values are as **p<0.01, *** p<0.001. (**c**) 4-dmH-suppressed tNOX expression was reversed by MG-132 in SAS cells and HSC-3 cells. Aliquots of cell lysates were resolved by SDS-PAGE and analyzed by western blotting. β-Actin was detected as an internal control. Values (mean ±SE)

*Figure 7 continued on next page*

*Figure 7 continued*

are from three independent experiments (n=3). The significance of differences between control and treatment groups was calculated using a one-way ANOVA followed by an appropriate post-hoc test such as LSD. The significant values are as *p<0.05, **p<0.01, *** p<0.001. (**d**) The lysates of HA-Ub overexpressing cells were immunoprecipitated with nonimmune IgG or an antibody against ENOX2, and the bound proteins were detected by western blotting with anti-HA or anti-ENOX2 antibodies. The total lysates were also immunoblotted with anti-tNOX or anti-β-actin antibodies. (**e**) The RNA interference-mediated tNOX depletion was conducted in SAS cells for 48 hr. The percentage of apoptotic/autophagic cells was examined by flow cytometry. The presented values (mean ± SDs) represent three independent experiments (n=3). The significance of differences between control and treatment groups was calculated using a one-way ANOVA followed by an appropriate post-hoc test such as LSD. The significant value is as ** p<0.01.

The online version of this article includes the following source data for figure 7:

**Source data 1.** Original files for the western blot analysis in *Figure 7*.

**Source data 2.** PDF containing *Figure 7b-e* of the relevant western blot analysis with the uncropped gels or blots with the relevant bands.

**Source data 3.** The flow cytometry dot blot autophagy and apoptosis data in *Figure 7e*.

SIRT1 and inhibited its activity and expression, to attenuate the growth of oral cancer cells. Given that SIRT1 contributes to a complex regulatory network, it is of primary importance to understand the fundamental regulatory mechanisms of SIRT1 in cell death regulation (*Wang et al., 2023*; *Wang et al., 2021a*; *Lee et al., 2018*; *Zhu et al., 2020*; *Ran et al., 2023*; *Zhang et al., 2020*).

Being an NAD$^+$-dependent protein deacetylase, it is not surprising that SIRT1 activity is primarily governed by NAD$^+$/NADH ratio (*Mouchiroud et al., 2013*; *He et al., 2022*; *Donmez, 2012*; *Teertam and Phanithi, 2022*; *Li et al., 2023*; *Bai et al., 2011*; *Ma et al., 2015*; *Fulco et al., 2003*; *Yang et al., 2022*). We further investigated the molecular events underlying the prominent inhibition of intracellular NAD$^+$/NADH ratio by 4-dmH, but not heliomycin. tNOX functions as a terminal hydroquinone oxidase, catalyzing not simply the oxidation of the reduced form of NADH but also the transfer of protons and electrons to molecular oxygen across the plasma membrane (*Hostetler et al., 2009*). tNOX exhibits relatively low expression in non-transformed cells, but is highly correlated with the hallmarks of cancer cells; positively regulates cell proliferation, and its downregulation is associated with cell-death-related pathways (*Hostetler et al., 2009*; *Sumiyoshi et al., 2022*; *Liu et al., 2008*; *Ronconi et al., 2016*). In recent years, efforts to target tNOX with RNA interference or conventional anticancer drugs have shown promising results in reducing cancer, not just in cell-based functional studies but also in animals inoculated with cancer cells of colon cancer and melanoma (*Liu et al., 2012*; *Islam et al., 2021*). Specifically, tNOX-depletion in cancer cells abolishes cancer phenotypes, reducing NAD$^+$ production, proliferation, and migration/invasion while increasing apoptosis (*Lee et al., 2015*; *Lin et al., 2016*; *Liu et al., 2008*; *Cheng et al., 2016*). Here, we report consistent findings in oral cancer cells, validating the essential role of tNOX in cell death regulation (*Figure 7e*). Moreover, tNOX-overexpressing in non-cancerous cells stimulates the growth of cells, decreases doubling time, and enhances cell migration (*Islam et al., 2019*; *Su et al., 2012*; *Zeng et al., 2012*; *Chueh et al., 2004*). We further substantiated the clinical relevance of tNOX in oral cancer by showing that the majority of our oral cancer patients exhibit higher tNOX protein expression in tumor tissues compared to their normal counterparts and there is a reverse correlation between tNOX expression and overall survival in a Kaplan-Meier analysis (*Figure 10*). Given that tNOX contributes to regulating SIRT1 activity, we propose that tNOX targeting is a better operational strategy for reducing cancer growth in both in vitro and in vivo studies.

Another significant finding of this study is the cellular thermal shift as analysis (CETSA)-based results showing the direct binding capacities of heliomycin and 4-dmH toward SIRT1 and tNOX. Moreover, 4-dmH and doxorubicin have very similar predicted affinities for tNOX protein. Based on our previous report, we postulated that the NADH-binding motif of tNOX is located within the range from Gly590 to Leu595 (*Chueh et al., 2002*). However, our simulation results indicated that the binding site of the studied compounds is located well away from the NADH-binding motif of the tNOX protein. The consistent interacting residues involved in the hydrogen bonds and hydrophobic interactions of the studied compounds were predicted to be Ile90, Lys98, Pro111, Pro113, Leu115, Pro117, and Pro118 (*Figure 9D*), which are not located in the NADH-binding motif. At this point, we are uncertain of the exact mechanism(s) by which these compounds inhibit tNOX. However, we speculate that the hydrophobic moieties and extra hydrogen bonds formed between 4-dmH or doxorubicin and the interacting residues of tNOX protein might affect its substrate binding. To clarify this issue, further efforts involving the co-crystallization of tNOX/inhibitor complexes are warranted.

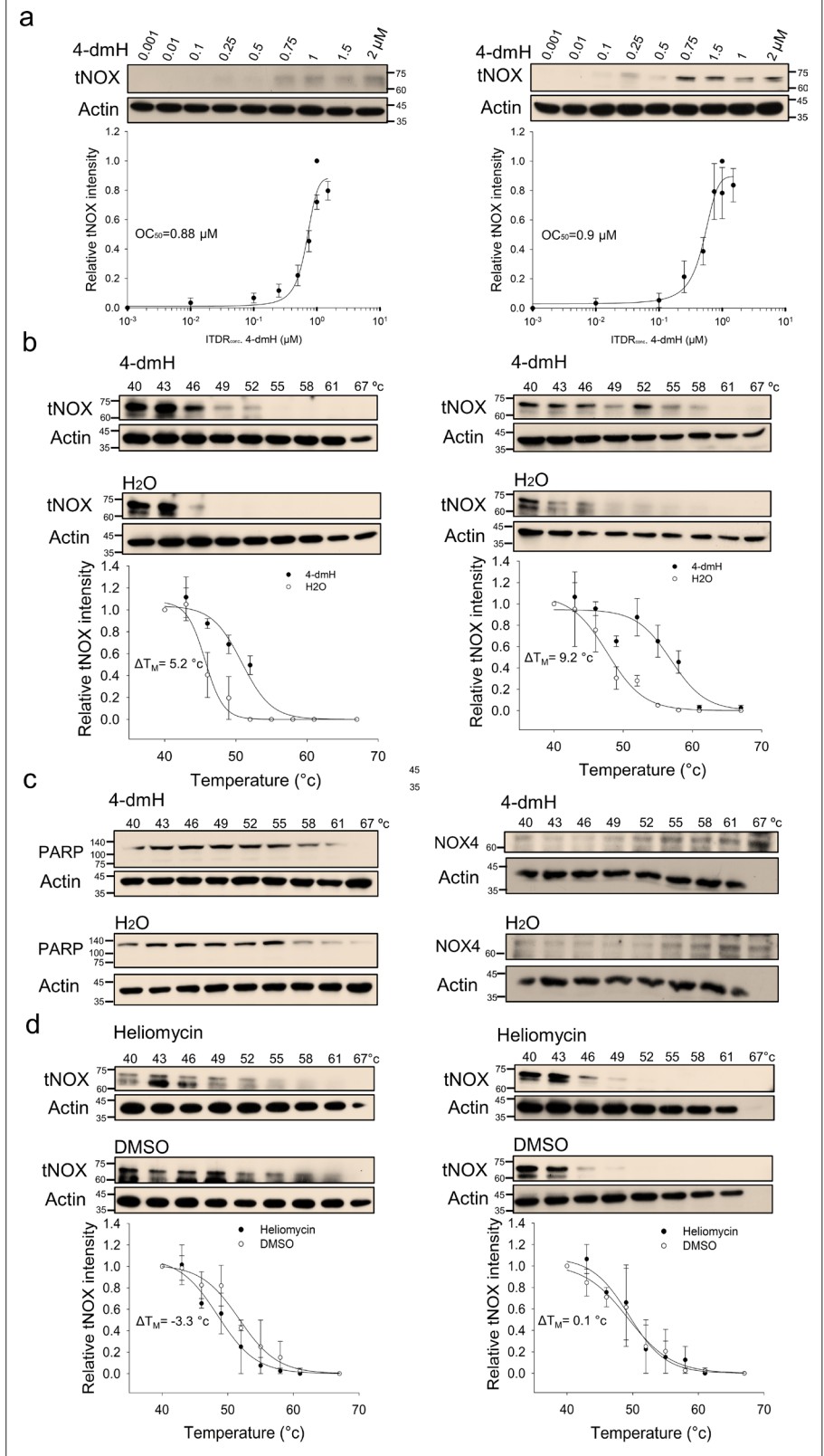

**Figure 8.** CETSA-based determination of binding between 4-dmH and tNOX protein. (**a**) Cells were incubated with different concentrations of the 4-dmH as described in the Methods. Dose-dependent thermal stabilization of tNOX was assessed after heating samples at 54 °C for 3 min in SAS cells and HSC-3 cells. The band intensities of tNOX were normalized with respect to the intensity of actin. Representative chemiluminescence data are shown on

*Figure 8 continued on next page*

*Figure 8 continued*

the top. Data are presented as the average and values (mean ±SE) are from three independent experiments (n=3). The representative images are generated by SigmaPlot followed by regression wizard. (**b**) CETSA-melting curves of tNOX in the presence and absence of 4-dmH in SAS cells and HSC-3 cells as described in the Methods. The immunoblot intensity was normalized to the intensity of the 40 °C samples. Representative chemiluminescence data are shown on the top. Data are presented as the average and values (mean ±SE) are from three independent experiments (n=3). The representative images are generated by SigmaPlot followed by regression wizard. The denaturation midpoints were determined using a standard process. (**c**) Western blot analysis for PARP or NOX4 in the presence and absence of 4-dmH in SAS cells are shown. The immunoblot intensity was normalized to the intensity of the 40 °C samples. (**d**) CETSA-melting curves of tNOX in the presence and absence of heliomycin in SAS cells and HSC-3 cells as described in the Methods. The immunoblot intensity was normalized to the intensity of the 40 °C samples. Representative chemiluminescence data are shown on the top. Data are presented as the average and values (mean ±SE) are from three independent experiments (n=3). The representative images are generated by SigmaPlot followed by regression wizard. The denaturation midpoints were determined using a standard process.

The online version of this article includes the following source data for figure 8:

**Source data 1.** Original file for the western blot analysis in *Figure 8*.

**Source data 2.** PDF containing *Figure 8* of the relevant western blot analysis with the uncropped gels or blots with the relevant bands.

Taken together, our present results show that the superior anticancer activity of 4-dmH over its parental heliomycin resides in its direct targeting of tNOX and SIRT1, and suggest that the tNOX-$NAD^+$-SIRT1 regulatory axis contributes to inducing apoptosis independent of p53 status, as evidenced by the results of in vitro and in vivo studies of oral cancer.

## Conclusions

In conclusion, we focused on the identification and validation of the intracellular targets of heliomycin and its water-soluble derivative, 4-dmH, by CETSA and molecular docking simulations in p53-functional SAS and p53-mutated HSC-3 oral cancer cells. Our CETSA- and cell-based results suggested that the targeting and inhibition of SIRT1 by heliomycin provoked autophagy and attenuated the growth of oral cancer cells. In addition to targeting SIRT1, as does its parental heliomycin, the enhanced water solubility of 4-dmH can maximize its anticancer effects and inhibit cooperatively tNOX and SIRT1 to induce apoptosis in oral cancer cells independent of their p53 status. Considering that tNOX not only supports but also exerts functions independent of SIRT1, the tNOX- and SIRT1-inhibiting function of 4-dmH, thus, results in the different biological outcomes from the SIRT1-binding heliomycin, as evidenced by the results of in vitro and in vivo studies of oral cancer.

## Methods
### Materials
The anti-SIRT1 (# 2496), anti-Atg5 (# 2630), anti-ULK1 (# 6439), anti-LC3 (# 4108), anti-PARP (# 9542), anti-Bak (# 12105), anti-Bax (# 2772), anti-Puma (# 4976), anti-Noxa (# 14766), anti-Bcl2 (# 15071), anti-c-Flip (# 56343), anti-c-Myc (# 5605), HA-tag (# 3724), and anti-cleaved caspase-3 (# 9661) antibodies were purchased from Cell Signaling Technology, Inc (Beverly, MA, USA). The anti-acetyl-c-Myc (# ABE26) antibody was from Millipore Corp. (Temecula, CA, USA). The anti-Atg7 (# NB110-55474) antibody was obtained from Novus Biologicals (Centennial, CO, USA). The anti-NOX4 (# SC-30141) antibody was obtained from Santa Cruz Biotechnology (Dallas, TX, USA). The antisera to tNOX used for immunoblotting were generated as described previously (***Chen et al., 2006***). The commercially available anti-ENOX2 (# 10423–1-AP) antibody and anti-β-actin (# 60008–1-Ig) antibodies were from Proteintech (Rosemont, IL, USA) was used for immunoprecipitation. The anti-mouse (# 115-035-003) and anti-rabbit IgG (# 111-035-003) antibodies were purchased from the Jackson ImmunoResearch Laboratories Inc, (West Grove, PA, USA).

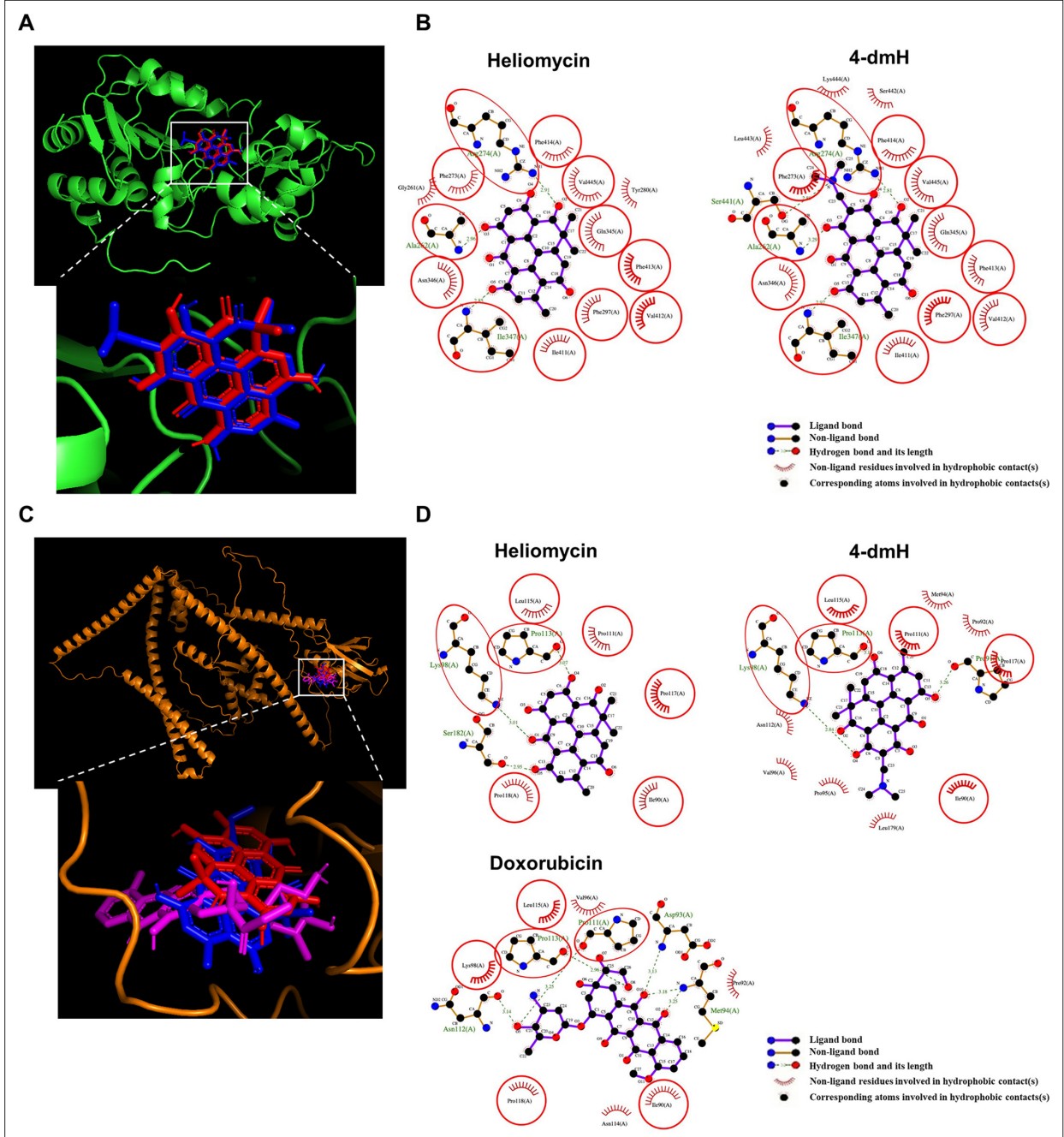

**Figure 9.** The binding modes of heliomycin and 4-dmH after docking into the pocket of SIRT1 (**a, b**) and tNOX (**c, d**). (**a**) Superimposition of the docked heliomycin (red) and 4-dmH (blue). (**b**) Schematic presentations of possible interactions between test compounds and SIRT1 residues. (**c**) Superimposition of the docked heliomycin (red) and 4-dmH (blue), and doxorubicin (purple). (**d**) Schematic presentations of possible interactions between test compounds and tNOX residues. The key residues surrounding the binding pocket of SIRT1 and tNOX were identified via the best docking pose of each compound. The red circles and ellipses indicate the identical residues that interacted with different compounds.

The online version of this article includes the following source data and figure supplement(s) for figure 9:

**Figure supplement 1.** The seven interaction residues on tNOX were substituted with alanine or glycine amino acids and then simulated the protein structures.

**Figure supplement 1—source data 1.** The simulated tNOX structures (**a, b**) and the binding modes of 4-dmH after docking study (**c, d**).

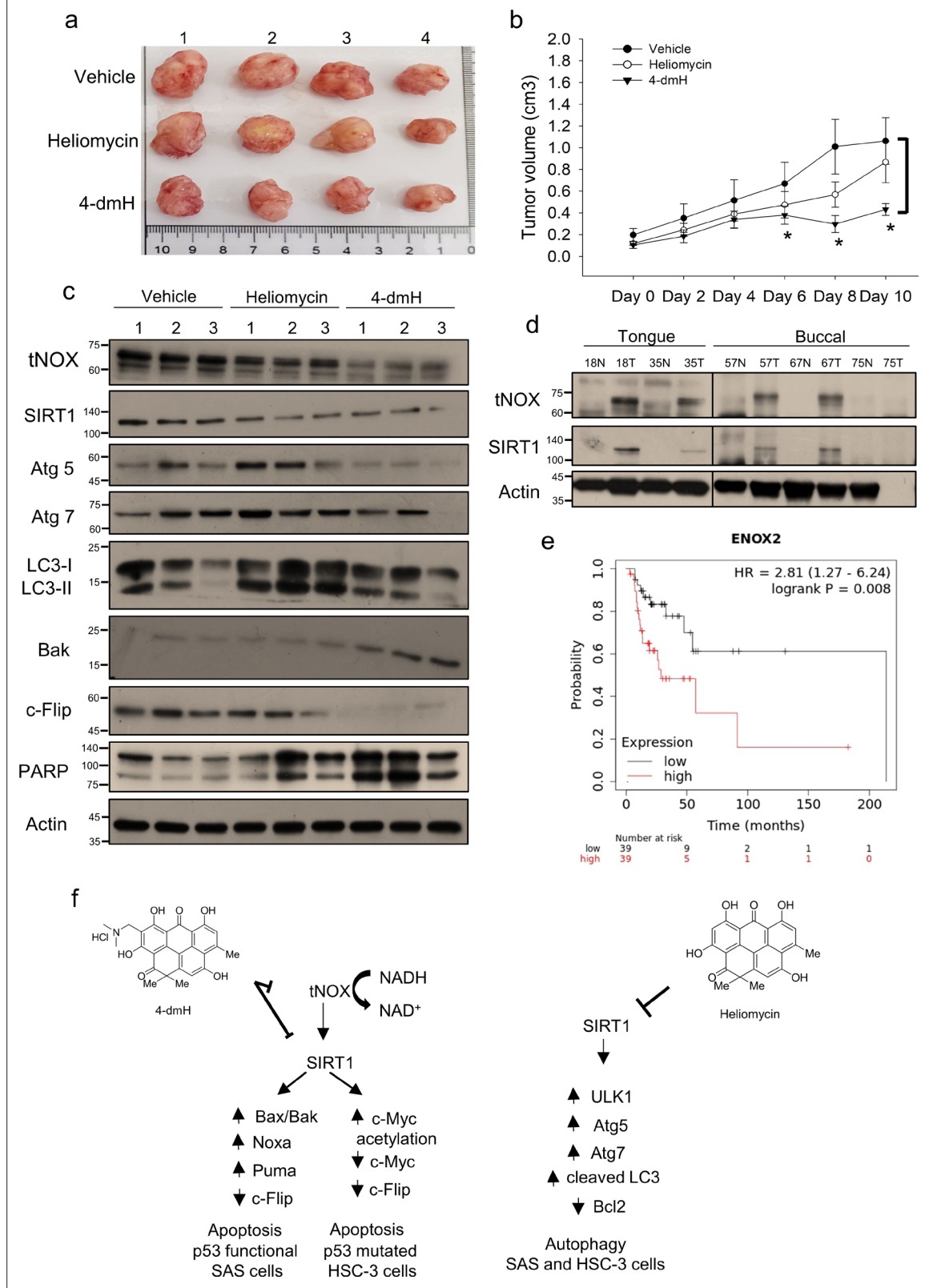

**Figure 10.** Therapeutic potential of heliomycin and 4-dmH in oral cancer. (**a–c**) In a tumor-bearing mouse xenograft model, control mice were intratumorally injected with vehicle buffer and treatment group mice were intratumorally treated with heliomycin or 4-dmH as described in the Methods. The morphology of the tumor tissues excised from tumor-bearing mice (**a**) and quantitative analysis of xenografted tumor volume during the treatment period (**b**) are shown. The significance of differences between control and treatment groups was calculated using a one-way ANOVA followed by an

*Figure 10 continued on next page*

*Figure 10 continued*

appropriate post-hoc test such as LSD. The significant value is as **p<0.05. (**c**) Tissues from three sets of tumor-bearing mice were grounded and prepared for western blotting analysis. (**d**) The tumor and adjacent tissues from oral cancer patients were grounded and prepared for western blotting analysis. β-actin was used as an internal loading control to monitor for equal loading. (**e**) Kaplan-Meier plots of the association between tNOX (ENOX2) expression and overall survival in 78 stage-III head-neck cancer patients. (**f**) Schematic diagram of the mechanism governing 4-dmH-induced apoptosis and heliomycin-mediated autophagy in oral cancer cells.

The online version of this article includes the following source data for figure 10:

**Source data 1.** Original files for the western blot analysis in *Figure 10*.

**Source data 2.** PDF containing *Figure 10c, d* of the relevant western blot analysis with the uncropped gels or blots with the relevant bands.

## Cell culture

The SAS (human squamous cell carcinoma of the tongue, JCRB0260) and HSC-3 (human tongue squamous cell carcinoma, JCRB0623) cells were purchased from the Japanese Collection of Research Bioresources Cell Bank (Osaka, Japan). The BEAS2B (human bronchial epithelial, ATCC CRL-9609) cells were obtained from the Bioresource Collection and Research Center, Hsinchu, Taiwan. Cells were grown in Dulbecco's Modified Eagle Medium (DMEM) and the media were supplemented with 10% FBS, 100 units/ml penicillin, and 50 µg/ml streptomycin. Cells were maintained at 37 °C in a humidified atmosphere of 5% $CO_2$ in the air, and the media were replaced every 2–3 days. Cells were treated with different concentrations of heliomycin (dissolved in DMSO) or the derivative 4-dmH (dissolved in water). All three cell lines were evaluated to be negative for mycoplasma using PCR-based assays (LookOut Mycoplama PCR Detection Kit, MP0040A, Merck KGaA, Darmstadt, Germany).

ON-TARGETplus tNOX siRNA and negative control siRNA were purchased from Thermo Scientific, Inc (Grand Island, NY). Briefly, cells were seeded in 10 cm dishes, allowed to attach overnight, and then transfected with tNOX siRNA or control siRNA using the Lipofectamine RNAiMAX Reagent (Gibco/BRL Life Technologies) according to the manufacturer's instructions.

## Chemistry

Heliomycin was produced at the Gause Institute of New Antibiotics (Moscow, Russian Federation) using *Actynomyces variabilis* var. *heliomycini* (*Brazhnikova et al., 1958*). 4-(Dimethylaminomethyl) heliomycin (4-dmH) was prepared from heliomycin by the previously reported method (*Nadysev et al., 2018*). The structures of both compounds are illustrated in *Figure 1*. The purities of the tested compounds of heliomycin and the water-soluble derivative were 97 and 99%, respectively, as examined by HPLC analysis.

## Continuous observation of cell proliferation by cell impedance determinations

For continuous monitoring of changes in cell proliferation, cells ($10^4$ cells/well) were plated onto E-plates, and the plates were incubated for 30 min at room temperature, and placed on an xCELLigence System (Roche, Mannheim, Germany). The cells were grown overnight and then exposed to heliomycin (dissolved in DMSO) or 4-dmH (dissolved in water), and impedance was measured every hour, as previously described (*Lee et al., 2015*). Cell impedance was defined by the cell index (CI) = $(Z_i − Z_0)$ [Ohm]/15[Ohm], where $Z_0$ is background resistance and $Z_i$ is the resistance at a given time point. A normalized cell index was determined as the cell index at a given time point ($CI_{ti}$) divided by the cell index at the normalization time point ($CI_{nml\_time}$).

## Measurement of mitochondrial membrane potential

Changes in mitochondrial membrane potential (a characteristic of apoptosis) were studied using JC-1 staining (Theremo Fisher Scientific Inc, Waltham MA, USA). Cells were treated with different concentrations of heliomycin or 4-dmH for 18 hr. The cells were then washed, incubated with 10 µM JC-1 at 37 °C for 30 min, washed with PBS, pelleted, and resuspended in PBS. The fluorescence intensity of cells was analyzed using a Beckman Coulter FC500 flow cytometer (Brea, CA, USA).

## Autophagy determination

Autophagosomes, which are acidic intracellular compartments that mediate the degradation of cytoplasmic materials during autophagy, were visualized by staining with acridine orange (AO; Sigma Chemical Co.). Cells were washed with PBS, stained with 2 mg/ml AO for 10 min at 37 °C, washed, trypsinized, and analyzed using a Beckman Coulter FC500 and cytoFLEX. The results are expressed as a percentage of total cells.

## Apoptosis determination

An Annexin V-FITC Apoptosis Detection Kit (BD Pharmingen, San Jose, CA, USA) was utilized to measure apoptosis. tNOX knockdown cells were cultured in 6 cm culture dishes, trypsinized, and harvested by centrifugation. Each pellet was rinsed with PBS, resuspended in 1×binding buffer, and stained with Annexin V-FITC (fluorescein isothiocyanate) followed by propidium iodide (PI; to determine necrotic or late apoptotic cells). The percentages of viable (FITC-negative and PI-negative), early apoptotic (FITC-positive and PI-negative), late apoptotic (FITC-positive and PI-positive), and necrotic (FITC-negative and PI-positive) cells were evaluated by cytoFLEX. The results are expressed as a percentage of total cells.

## Colony formation assay

Onto each 6 cm dish, $10^3$ cells were cultured in growth medium with various concentrations of heliomycin and 4-dmH at least 10 days. At the end of the experiment, colonies were fixed in 1.25% glutaraldehyde at room temperature for 30 min, washed with distilled water, and colored with a 0.05% methylene blue mixture. The number of colonies was determined and documented.

## Cell cycle analysis

In brief, after treatments, $10^6$ cells were collected and washed in PBS, slowly fixed in 75% ethanol, and kept at –20 °C for at least 1 hr. The cell pellet was then washed again with PBS, and centrifuged at $500 \times g$ for 5 min. The pellet was resuspended in 200 µl cold PBS and nuclear DNA was stained in the dark with propidium iodide (PI) solution (20 mM Tris pH 8.0, 1 mM NaCl, 0.1% NP-40, 1.4 mg/mL RNase A, 0.05 mg/mL PI) for 30 min on ice. Total cellular DNA content was analyzed with an FC500 flow cytometer (Beckman Coulter Inc, Brea, CA, USA).

## Cellular target identification by cellular thermal shift assay (CETSA)

CETSA was used to identify whether SIRT1 or tNOX are cellular targets of heliomycin and its water-soluble derivative, 4-dmH. Samples were prepared from control and compound-exposed cells. For each set, $2×10^7$ cells were seeded in a 10 cm culture dish. After 24 hr of culture, the cells were pretreated with 10 µM MG132 for 1 hr, washed with PBS, treated with trypsin, collected by centrifugation at 12,000 rpm for 3 min at room temperature, gently resuspended with 1 ml of PBS, pelleted by centrifugation at 7500 rpm for 3 min at room temperature, and resuspended with 1 ml of lysis buffer (20 mM Tris-HCl pH 7.4, 100 mM NaCl, 5 mM EDTA, 2 mM phenylmethylsulfonyl fluoride (PMSF), 10 ng/ml leupeptin, and 10 µg/ml aprotinin). The samples were transferred to Eppendorf tubes and subjected to three freeze-thaw cycles; for each cycle, they were exposed to liquid nitrogen for 3 min, placed in a heating block at 37 °C for 3 min, and vortexed briefly. For the experimental sample set, heliomycin or its derivative was added to a final concentration of 20 µM; the same volume of vehicle solvent was added for the control sample set. The samples were heated at 37 °C for 1 hr and dispensed to 100 µl aliquots. Pairs consisting of one control aliquot and one experimental aliquot were heated at 40 °C, 43 °C, 46 °C, 49 °C, 52 °C, 55 °C, 58 °C, 61 °C, or 67 °C for 3 min. Insoluble proteins were separated by centrifugation at 12,000 rpm for 30 min at 4 °C, and the supernatants (containing soluble proteins) were resolved by SDS-PAGE. Western blot analysis was performed using commercially available SIRT1 antibodies or tNOX antisera (*Liu et al., 2008*; *Chen et al., 2006*). β-Actin was detected as a loading control.

An isothermal dose-response fingerprint ($ITDRF_{CETSA}$) was obtained in a manner similar to that described above for the CETSA melting-curve experiments. Cells were seeded in 60 mm culture dishes. After 24 hr of culture, the cells were pretreated with 10 µM MG132, exposed to the test compound for 1 hr (final concentration, 0.001, 0.01, 0.1, 0.25, 0.5, 0.75, 1, 1.5, or 2 µM), washed with PBS, treated with trypsin, pelleted at 12,000 rpm for 5 min at room temperature, gently resuspended

with 1 ml of PBS, pelleted at 7500 rpm for 3 min at room temperature, and resuspended with the same described above. The samples were subjected to three freeze-thaw cycles, wherein they were exposed to liquid nitrogen for 3 min, placed in a heating block at 25 °C for 3 min, and vortexed briefly. The samples were then heated at 54 °C for 3 min and cooled for 3 min at room temperature. Insoluble proteins were separated by centrifugation at 12,000 rpm for 30 min at 4 °C, and the supernatants (containing soluble proteins) were subjected to SDS-PAGE and western blot analysis using an antibody against SIRT1 or antisera to tNOX. β-Actin was used as a loading control.

## Western blot analysis and immunoprecipitation

Cell extracts were prepared in the lysis buffer described above. Volumes of extract containing equal amounts of proteins (40 µg) were resolved by SDS-PAGE and transferred to PVDF membranes (Schleicher & Schuell, Keene, NH, USA), and the membranes were blocked, washed, and probed with the indicated primary antibody overnight. The membranes were washed, incubated with horseradish peroxidase-conjugated secondary antibody for 1 hr, and developed using enhanced chemiluminescence (ECL) reagents (Amersham Biosciences, Piscataway, NJ, USA) according to the manufacturer's protocol.

For immunoprecipitation, protein extracted from cells grown in 100 mm dishes were pre-cleared with 20 µl of Protein G Agarose Beads (for rabbit antibodies) for 1 hr at 4 °C with rotation. Ubiquitin antibodies or control IgG were incubated onto beads in 500 µl of lysis buffer, overnight with rotation at 4 °C. Beads were precipitated by centrifugation at 3000 rpm for 2 min at 4 °C. Beads were washed three times with lysis buffer and samples were prepared for western blotting analysis.

## Determination of SIRT1 deacetylase activity in vitro

SIRT1 deacetylase activity was determined using a SIRT1 Direct Fluorescent Screening Assay Kit (Cayman Chemical Company, Ann Arbor, MI) according to the manufacturer's protocol. Briefly, substrate solution was prepared by adding 240 µl of NAD$^+$ solution and 850 µl of diluted Assay Buffer to 15 µl of the p53 peptide Arg-His-Lys-Lys(εacetyl)-AMC, to yield a final concentration of 125 µM peptide (substrate) and 3 mM of NAD$^+$. Using a 96-well plate, the background was determined in wells containing 30 µl of Assay Buffer and 5 µl of DMSO. Maximal initial activity (defined as 100%) was determined in wells containing 5 µl of diluted human recombinant SIRT1, 25 µl of Assay Buffer, and 5 µl of DMSO. To measure the ability of the test compound to modulate SIRT1 activity, 5 µl of 2 µM compound was added to wells containing 25 µl of Assay Buffer and 5 µl of diluted human recombinant SIRT1. Reactions were initiated by adding 15 µl of Substrate Solution to each well. The plate was then covered and incubated on a shaker for 45 min at room temperature. Reactions were stopped by adding 50 µl of Stop/Developing Solution to each well and incubating the plate for 30 min at room temperature. Plates were read in a fluorimeter using an excitation wavelength of 350–360 nm and an emission wavelength of 450–465 nm.

## Measurement of the intracellular NAD$^+$/NADH ratio

The oxidized and reduced forms of intracellular NAD were determined using an NADH/NAD Quantification Kit (BioVision Inc Milpitas, CA, USA), as described by the manufacturer. Briefly, $2 \times 10^5$ cells were washed with cold PBS, pelleted, and extracted by two freeze/thaw cycles with 400 µl of NADH/NAD$^+$ extraction buffer. The samples were vortexed and centrifuged at 14,000 rpm for 5 min. The extracted NADH/NAD$^+$-containing supernatant (200 µl) was transferred to a microcentrifuge tube, heated to 60 °C for 30 min (to decompose NAD$^+$ but not NADH), and then placed on ice. The samples were then centrifuged and transferred to a multiwell plate. Standards and an NAD$^+$ cycling mix were prepared according to the manufacturer's protocol. The reaction mix was distributed at 100 µl/well to wells of a 96-well plate pre-loaded with NADH standards and samples, and the plate was incubated at room temperature for 5 min to convert NAD$^+$ to NADH. The provided NADH developer solution was added to each well, and plates were incubated at room temperature for 15 or 30 min. The reaction was stopped with 10 µl of stop solution per well, and absorbance was measured at 450 nm.

## Molecular docking simulation

A crystal structure of the SIRT1 catalytic domain bound to an EX-527 analog (PDB: 4I5I) (*Zhao et al., 2013*) was employed for our molecular docking study. EX-527 is a nanomolar SIRT1 inhibitor with an

IC$_{50}$ value as low as 38 nM (**Solomon et al., 2006**); therefore its co-crystal structure of SIRT1 protein was judged to be suitable for our docking analysis. The water molecule and ligand molecule of the initial crystal structure were removed using the PyMOL program (https://pymol.org/; accessed on 1 November 2022; PyMOL Molecular Graphics System, Schrodinger, New York, NY) to prepare for the docking analysis. A predicted full-length structure of the tNOX protein was obtained from the Alpha-Fold Protein Structure Database (**Jumper et al., 2021**; **Varadi et al., 2022**) and used in this study. Molecular docking was performed using the AutoDock Vina package (**Trott and Olson, 2010**) in the PyRx software (**Dallakyan and Olson, 2015**) to assess the probable binding modes of heliomycin and its derivative in the SIRT1 catalytic domain and the protein structure of tNOX. The docking site was determined according to the inhibitor binding site of the EX-527 analog in the crystal structure of SIRT1, which was used as the setting in the grid selection. Because we lacked information about the ligand-bound pocket, blind docking was performed by adopting the whole-protein structure of tNOX, as the grid selection. Each compound was optimized in its molecular geometry, torsional barriers, and intermolecular-interaction geometry using the MMFF94 forcefield in CHARMM (**Brooks et al., 2009**). The best docking conformation of each compound, which was chosen based on the lowest binding energy, was employed in the interaction analyses. The interaction diagram was generated using the LigPlot +software (**Laskowski and Swindells, 2011**) to display the hydrogen bonds and hydrophobic moieties. The docking pose of each compound was visualized using the PyMOL program.

### Homology modeling of tNOX protein structure

To evaluate the impact of key interacting residues derived from post-docking analysis, the key residues on the tNOX protein were substituted with alanine or glycine amino acid and then conducted the protein structure simulation. The tNOX structure from the AlphaFold database was used as a template during these processes. The 3D structures of the modified tNOX protein sequences were homology-modeled using the I-TASSER standalone package version 5.2 (**Yang et al., 2015**) (https://zhanggroup.org/I-TASSER/download/) with default parameters. The simulated structures were also applied in the molecular docking study to assess possible binding affinity and binding modes between 4-dmH and the modified tNOX protein.

### In vivo xenograft studies

Specific pathogen-free (SPF) ASID mice were purchased from the National Laboratory Animal Center (Taipei, Taiwan). The animal use protocols were approved by the Institutional Animal Care and Use Committee of National Chung Hsing University (Taichung, Taiwan) (IACUC No. 108–127). Mice were subcutaneously inoculated with 100 µl containing $2 \times 10^6$ live SAS cells in PBS. The tumor-bearing animals were randomized divided into three groups (n=4 per group) when the tumor mass reached an average diameter of 0.5–1 cm. Mice were intratumorally injected with vehicle buffer (20% DMSO in PBS) as untreated control (Vehicle group), with 200 µg heliomycin in vehicle buffer (Heliomycin group), with 200 µg 4-dmH in vehicle buffer (4-dmH group). Intratumoral therapy was performed three times at 1 week intervals. The tumor size was recorded every 2 days and the tumor volume was calculated using the formula: length $\times$ width$^2 \times 0.5$. Mice were euthanized 4 days after the final treatment. The significance of differences in tumor size was determined by a one-way ANOVA.

### Patient specimens

Five pairs of cancer tissues and their corresponding normal counterparts were obtained from Changhua Christian Hospital in Taiwan (CCH IRB No. 130616). Diagnosis of oral cancer was based on histological examination of hematoxylin and eosin-stained tissue section. Patient specimens were immediately frozen in liquid nitrogen after surgery and used for western blot analysis.

### Statistics

All data are expressed as the means ±SEs/SDs of three independent experiments. Between-group comparisons were performed using a one-way analysis of variance (ANOVA) followed by an appropriate posthoc test. A value of $p < 0.05$ was considered to be statistically significant.

## Acknowledgements

Financial support was provided by grants from the Ministry of Sciences and Technology, Taiwan (MOST 106–2320-B-005–008-MY3 to PJC and MOST 108–2923-B-005–001-MY3 to PJC), the Russian Foundation for Basic Research (Project 19-53-52008 to AES), and the National Chung Hsing University and Changhua Christian Hospital (NCHU-CCH 11105 to CWW).

## Additional information

### Funding

| Funder | Grant reference number | Author |
|---|---|---|
| The Ministry of Sciences and Technology, Taiwan | MOST 106-2320-B-005-008-MY3 | Pin Ju Chueh |
| The Minsitry of Sciences and Technology, Taiwan | MOST 108-2923-B-005-001-MY3 | Pin Ju Chueh |
| Russian Foundation for Basic Research | Project 19-53-52008 | Andrey E Shchekotikhin |
| National Chung Hsing University and Changhua Christian Hospital | NCHU-CCH 11105 | Che-Wei Wang |

The funders had no role in study design, data collection and interpretation, or the decision to submit the work for publication.

### Author contributions

Atikul Islam, Data curation, Formal analysis, Validation, Investigation, Methodology, Writing – original draft; Yu-Chun Chang, Data curation, Formal analysis, Validation, Investigation, Methodology; Xiao-Chi Chen, Data curation, Validation, Investigation, Visualization, Methodology; Chia-Wei Weng, Data curation, Software, Formal analysis, Visualization, Methodology, Writing – original draft; Chien-Yu Chen, Software, Formal analysis, Validation, Investigation; Che-Wei Wang, Resources, Formal analysis, Funding acquisition, Investigation, Methodology, Writing – original draft; Mu-Kuan Chen, Conceptualization, Resources, Supervision, Funding acquisition, Writing – review and editing; Alexander S Tikhomirov, Conceptualization, Resources, Data curation, Funding acquisition, Validation, Writing – original draft, Writing – review and editing; Andrey E Shchekotikhin, Conceptualization, Resources, Data curation, Funding acquisition, Writing – review and editing; Pin Ju Chueh, Conceptualization, Resources, Supervision, Funding acquisition, Investigation, Writing – original draft, Project administration, Writing – review and editing

### Author ORCIDs

Atikul Islam ⬤ http://orcid.org/0009-0004-9575-5963
Chia-Wei Weng ⬤ http://orcid.org/0000-0002-4842-7308
Pin Ju Chueh ⬤ http://orcid.org/0000-0002-3200-7552

### Ethics

Specific pathogen-free (SPF) ASID mice were purchased from the National Laboratory Animal Center (Taipei, Taiwan). The animal use protocols were approved by the Institutional Animal Care and Use Committee of National Chung Hsing University (Taichung, Taiwan) (IACUC No. 108-127).

Joint Public Review: https://doi.org/10.7554/eLife.87873.3.sa1
Author Response https://doi.org/10.7554/eLife.87873.3.sa2

## Additional files

### Supplementary files
• MDAR checklist

## Data availability

All data generated and analyzed during this study are included in the manuscript and supporting files.

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
