## [Editor Report · eLife assessment]

This **useful** study reports that a water-soluble analog of heliomycin, 4-dmH, induces protein degradation of not only SirT1 but also tNOX, unlike heliomycin, which induces degradation of SirT1 but not tNOX, a difference that could in principle explain why 4-dmH induces apoptosis while heliomycin induces autophagy. The presented data provide **solid** support for the authors' conclusions.

---

## [Referee Report · Joint Public Review]

Previous findings by authors show that heliomycin induces autophagy to inhibit cancer progression, while its water-soluble analogs induce apoptosis. Here, they show that one of the analogs, 4-dmH, binds to tNOX, a NADH oxidase which supports SirT1 activity, in addition to SirT1, while heliomycin only binds to SirtT1 but not tNOX, using CETSA and in silico molecular docking studies, in human oral cancer cells. The additional binding activity of 4-dmH to tNOX might explain the different biological outcome from heliomycin. 4-dmH induces ubiquitination and degradation of tNOX protein, in dependent of p53 status. The tumor suppressive effect of 4-dmH (by intra-tumoral injections) is better than heliomycin. TCGA data base analysis suggests that high tNOX mRNA expression is correlated with poor prognosis of oral cancer patients.

This group has been a leading lab of chemical and biological characterization of heliomycin and its analogs. Their findings are interesting and advance their previous findings. The revised manuscript well responded to the reviewers' concerns.

---

## [Author Response]

The following is the authors’ response to the original reviews.

Responses to reviewers’ comments

(1) The rationale of selecting tNOX/ENOX2 as a potential target of 4-dmH, but not heliomycin, is unclear by taking a biased approach. Thus, there is high possibility that 4-dmH binds to other proteins involved in apoptosis inhibition. An unbiased screen to identify 4-dmH-binding proteins would be a better approach unless there is a clear and logical rationale.

We apologize for this oversight. In response to this comment, we rewrote the abstract, reorganized the results, and added more references to better introduce tNOX/ENOX2.

A) Under the “4-dmH, but not heliomycin, targets intracellular tNOX, an upstream regulator of SIRT1” result section:

We next addressed the molecular mechanisms underlying SIRT1 inhibition and concurrent cell death by these two compounds in oral cancer cells. Being an NAD+-dependent protein deacetylase, SIRT1 activity is primarily governed by NAD+/NADH ratio, thus, there exists a positive correlation between these two [1-9]. We then questioned whether these two compounds inhibit SIRT1 by affecting the intracellular NAD+/NADH levels, and were surprised to find that 4-dmH, but not heliomycin, caused a prominent inhibition of intracellular NAD+/NADH ratio (revised Fig. 7a). The discrepancy in their ability to reduce NAD+ generation led us to explore the role of a tumor-associated NADH oxidase (tNOX, ENOX2) in 4-dmH-suppressed SIRT1 and apoptosis induction. We have previously reported that tNOX inhibition reduced the intracellular NAD+/NADH ratio and SIRT1 deacetylase activity, increasing p53 acetylation and apoptosis [10-13]. In the light of this information, we assessed the effect of the compounds on tNOX expression and found that 4-dmH, but not heliomycin, considerably diminished tNOX protein expression in a concentration-dependent manner (Fig. 7b).

B) To demonstrate that our results from ligand-binding assays (CETSA) were specific to tNOX, we conducted more CETSA experiments to exclude PARP or NOX4 targets of 4-dmH. PARP acts as a DNA damage sensor and also a NAD+-consuming enzyme, affecting many cellular functions [14]. NOX4 belongs to the NOX family of NADPH oxidases that mediate electron transport through intracellular membranes and is also shown to be involved in tumorigenesis [15, 16]. We show that 4-dmH treatments did not seem to increase the melting temperature of PARP or NOX4, excluding those two proteins as potential targets of 4-dmH (revised Fig. 8c).

**Author response image 1. sa2fig1:** 

(2) The authors should show whether heliomycin indeed does not induce apoptosis, while 4-dmH cannot induce autophagy.

We have reorganized and revised our manuscript and figures (Fig. 5 and Fig. 6) to better demonstrate the different cell death pathways associated with heliomycin and 4-dmH. Using flow cytometry, we show that heliomycin, but not 4-dmH, induced autophagy in two lines of oral cancer cells (Fig. 5a). In the revision, we moved up the analysis of apoptosis by JC-1 staining to Figure 5 (revised Fig. 5b). We also reorganized the protein analysis to demonstrate better the downregulation of pro-apoptotic Bak and Puma and a lack of caspase 3-directed PARP cleavage, indicating the ineffective apoptosis by heliomycin (revised Fig. 5c). Similarly, we found that the absence of upregulation of ULK1, Atg 5, Atg7, and cleaved LC3-II provided evidence for the inadequate autophagy by 4-dmH (revised Fig. 5d). Attached please see the revised Figure 5.

(3) They should demonstrate whether genetic knockdown of tNOX, SirT1, or both tNOX and SirT1 induces apoptosis or autophagy and also reduces malignant properties of oral cancer cells.

A) In the revision, we conducted more experiments utilizing the RNAi-knockdown to understand the role of tNOX on the regulation of apoptosis or autophagy. Our results indicate that the tNOX-depletion effectively provoked spontaneous apoptosis and autophagy in SAS cells (revised Fig. 7e). However, given that SIRT1 per se is not the focus of this present study and SIRT1-knockdown has been shown to increase apoptotic population by other groups [17] [18], we decided not to pursue it further.

**Author response image 3. sa2fig3:** 

B) In our earlier studies, we have adequately demonstrated that tNOX confers a survival advantage for cancer cells. For example, tNOX-deficiency by RNA interference in cancer cells abolishes cancer phenotypes, reducing NAD+ production, proliferation, and migration/invasion while increasing apoptosis [19-22]. On the other hand, tNOX-overexpressing in non-cancerous cells stimulates the growth of cells, decreases doubling time, and enhances cell migration [23-26].

(4) The authors should examine whether overexpression of SirT1 or tNOX in cells treated with heliomycin or 4-dmH could nullify heliomycin-induced autophagy and 4-dmH-induced apoptosis. Also, instead of overexpressing tNOX, they can supplement NAD into cells treated with 4-dmH.

A) The utilization of tNOX overexpression has been previously reported in several studies, demonstrating that tNOX-overexpressing in non-cancerous cells stimulates the growth of cells, decreases doubling time, and enhances cell migration [23-26]. However, in our experiences, the effect of tNOX overexpression in cancer cells is much less apparent than that in non-cancerous cells. Thus, we decided not to study it further, given that our results from tNOX knockdown have evidently signified the role of tNOX in the regulation of apoptosis and autophagy.

B) Since SIRT1 is not the major focus of this present study and SIRT1-overexpression has been shown to reduce stress-mediated apoptosis by other groups [27, 28], we decided not to pursue it further.

C) The systemic deterioration in NAD+ level has been correlated with many diseases and aging [29-31]. In this regard, NAD+ administration was reported to attenuate doxorubicin-induced apoptosis in the liver of mice, suggesting a protective effect [32]. The administration of nicotinamide riboside (NR), a precursor of NAD+, was also demonstrated to prevent ROS generation and apoptosis in the mouse sepsis models [33]. With data from these animal studies already demonstrating the benefits of NAD+ supplements, we decided not to conduct similar experiments in a cell-based setting.

(5) Related to Fig. 5C and 6a, the authors should examine the effects of heliomycin and 4-dmH on the cell cycle profiles, Annexin V positivity, and colony formation.

We added the results from colony-forming assays and revealed that both compounds exhibited high growth-suppressive ability against oral cancer cells (revised Fig. 6c). Nevertheless, we showed that the diminution in growth by the compounds was least likely to arise from cell cycle arrest mediated by these two compounds (revised Fig. 6d). Due to the possible interference of the fluorescence wavelength of heliomycin/derivative, we examined JC-1 staining rather than Annexin V positivity. The apoptotic effect of the compounds was demonstrated in revised Fig. 5b in the revision.

**Author response image 4. sa2fig4:** 

(6) They should also examine whether either or both heliomycin and 4-dmH induce reactive oxygen species (ROS).

In our previous report, we examined the effects of heliomycin and 4-dmH on oxidative stress utilizing H2DCFDA [34]. The dye fluoresces in the presence of intracellularly generated reactive oxygen species (ROS). We showed that 4-dmH significantly induced the generation of ROS generation. However, no marked ROS generation was observed in cells exposed to heliomycin.

(7) Related to Fig. 9d, they should mutate amino acid residue(s) in tNOX that are crucial for the 4-dmH-tNOX binding, including Ile 90, Lys98, Pro111, Pro113, Leu115, Pro117, and Pro118, to examine whether these mutants lose the binding to 4-dmH and fail to rescue 4-dmH-induced apoptosis, unlike wild-type tNOX.

For further evaluation of the importance of the consistent interaction residues in the three docked compound-tNOX complexes, the seven interaction residues on tNOX were substituted with alanine or glycine amino acids and then simulated the protein structures. The simulated protein structures appear slightly different from the original tNOX structure. Overall, the root mean square difference between the original tNOX structure and the structures with residues substituted by alanine or glycine amino acids was estimated at 3.339 or 4.024 angstroms (Å), respectively (Fig. S1a). The simulated protein structures were also employed to conduct the docking analysis for 4-dmH. The results of further docking analysis revealed that 4-dmH could bind within the same pocket of different types of tNOX structures but with varying orientations (Fig. S1b). This observation also suggests that the replacement of both key residues with alanine or glycine could result in a reduction of the binding affinity of 4-dmH to tNOX, with values of -8.2 and -7.6 kcal/mol, respectively. Moreover, the substitution of both key residues with alanine or glycine also reduces the number of the original interacting residues and interaction forces in core moieties in the 4-dmH-tNOX complexes (Fig. S1c and S1d). Together, our experimental results and molecular docking simulations are consistent with the notion that 4-dmH possesses a better affinity ability for tNOX than for SIRT1.

**Author response image 5. sa2fig5:** The simulated tNOX structures (a, b) and the binding modes of 4-dmH after docking study (c, d). (a) Superimposition of three types of tNOX structures, including the original tNOX structure (orange) and the critical residues in tNOX protein substituted with alanine (magenta) or glycine (cyan). The substituted residues were shown as sticks. (b) Superimposition of the docked 4-dmH (blue). (c) Schematic presentations of possible interactions between 4-dmH and the interacted residues in tNOX protein substituted with alanine. (d) Schematic presentations of possible interactions between 4-dmH and the interacted residues in tNOX protein substituted with glycine. The key residues were identified based on the best docking pose of 4-dmH. The red circles and ellipses indicate the identical residues that interacted with different types of tNOX structures.

(8) Related to Fig. 10a, heliomycin appears to also reduce tNOX levels (although the extent is not as robust as 4-dmH), which is not expected since heliomycin does not bind to tNOX. They should compare the effects of heliomycin and 4-dmH on reducing the protein levels of tNOX. If heliomycin does not change the tNOX protein levels, then they need to discuss why heliomycin reduces tNOX levels in vivo.

In our previous studies, we have shown that tNOX knockdown partially attenuates SIRT1 expression and represses growth in various cancer cell types, such as lung [22], bladder [20], and stomach [13]. We also observed that tNOX is acetylated/ubiquitinated under certain stresses and SIRT1 depletion affects tNOX expression (data not shown). It is speculated that SIRT1 deacetylates tNOX and modulates its protein stability. Thus, there is a reciprocal regulation between tNOX and SIRT1. Although heliomycin does not bind to tNOX, its inhibition of SIRT1 activity/expression might also have an impact on tNOX expression.

(9) Related to Fig. 10F, if tNOX is an upstream regulator of SirT1 and both heliomycin and 4-dmH ultimately target SirT1, it is unclear why heliomycin and 4-dmH cause different biological outcomes. One explanation is that tNOX has apoptosis-inhibiting function other than supporting (or independent of) SirT1 and hence 4-dmH-mediated tNOX inhibition causes apoptosis rather than autophagy. They should explain and discuss more about whether tNOX-inhibiting/binding function of 4-dmH is sufficient to explain the different biological outcomes from heliomycin.

Thank you for this valuable suggestion. Indeed, in our earlier studies, we have adequately demonstrated that tNOX-deficiency by RNA interference in cancer cells abolishes cancer phenotypes, reducing NAD+ production, proliferation, and migration/invasion while increasing apoptosis; thus, tNOX confers a survival advantage for cancer cells [19-22]. On the other hand, tNOX-overexpressing in non-cancerous cells stimulates the growth of cells, decreases doubling time, and enhances cell migration [23-26]. With these lines of evidence, we believe that tNOX not only supports but also exerts functions independent of SIRT1. The tNOX- and SIRT1-inhibiting function of 4-dmH, thus, results in the different biological outcomes from the SIRT1-binding heliomycin.

(10) They should examine the effects of heliomycin and 4-dmH on cell viability of non-tumor cells to examine their toxicities.

Using cell impedance measurements, we also examined the effects of heliomycin and 4-dmH on the proliferation of human non-cancerous BEAS-2B cells. Our results demonstrated that heliomycin did not exhibit cytotoxicity toward human non-cancerous BEAS-2B cells (revised Fig. 6a). Furthermore, the water-soluble 4-dmH effectively diminished cell proliferation in a dose-dependent manner in oral cancer cells, but much less apparent in that of BEAS-2B cells (revised Fig. 6b). Similar results were reported in our previous study, indicating that 4-dmH displayed much higher IC50 values against non-cancerous human dermal microvascular endothelium HMEC-1 cells compared to those of tumor cells [34].

**Author response image 6. sa2fig6:** 

(11) They should consistently use either tNOX or ENOX2 to avoid confusion.

Thank you for the suggestion. We have now consistently used tNOX throughout the manuscript. However, for the revised Figure 7d, the commercially available antibody to ENOX2 (from Proteintech, Rosemont, IL, USA) is different from the one to tNOX (produced in our laboratory) and this is the only place we have used ENOX2 rather than tNOX.

References

1. Mouchiroud L, Houtkooper RH, Moullan N, Katsyuba E, Ryu D, Canto C, Mottis A, Jo YS, Viswanathan M, Schoonjans K et al: The NAD(+)/Sirtuin Pathway Modulates Longevity through Activation of Mitochondrial UPR and FOXO Signaling. Cell 2013, 154(2):430-441.

2. He S, Gao Q, Wu X, Shi J, Zhang Y, Yang J, Li X, Du S, Zhang Y, Yu J: NAD(+) ameliorates endotoxin-induced acute kidney injury in a sirtuin1-dependent manner via GSK-3beta/Nrf2 signalling pathway. J Cell Mol Med 2022, 26(7):1979-1993.

3. Donmez G: The neurobiology of sirtuins and their role in neurodegeneration. Trends Pharmacol Sci 2012, 33(9):494-501.

4. Teertam SK, Phanithi PB: Up-regulation of Sirtuin-1/autophagy signaling in human cerebral ischemia: possible role in caspase-3 mediated apoptosis. Heliyon 2022, 8(12):e12278.

5. Li BY, Peng WQ, Liu Y, Guo L, Tang QQ: HIGD1A links SIRT1 activity to adipose browning by inhibiting the ROS/DNA damage pathway. Cell reports 2023, 42(7):112731.

6. Bai P, Canto C, Oudart H, Brunyanszki A, Cen Y, Thomas C, Yamamoto H, Huber A, Kiss B, Houtkooper RH et al: PARP-1 inhibition increases mitochondrial metabolism through SIRT1 activation. Cell Metab 2011, 13(4):461-468.

7. Ma Y, Nie H, Chen H, Li J, Hong Y, Wang B, Wang C, Zhang J, Cao W, Zhang M et al: NAD(+)/NADH metabolism and NAD(+)-dependent enzymes in cell death and ischemic brain injury: current advances and therapeutic implications. Curr Med Chem 2015, 22(10):1239-1247.

8. Fulco M, Schiltz RL, Iezzi S, King MT, Zhao P, Kashiwaya Y, Hoffman E, Veech RL, Sartorelli V: Sir2 regulates skeletal muscle differentiation as a potential sensor of the redox state. Mol Cell 2003, 12(1):51-62.

9. Yang Y, Liu Y, Wang Y, Chao Y, Zhang J, Jia Y, Tie J, Hu D: Regulation of SIRT1 and Its Roles in Inflammation. Front Immunol 2022, 13:831168.

10. Tikhomirov AS, Shchekotikhin AE, Lee YH, Chen YA, Yeh CA, Tatarskiy VV, Jr., Dezhenkova LG, Glazunova VA, Balzarini J, Shtil AA et al: Synthesis and Characterization of 4,11-Diaminoanthra[2,3-b]furan-5,10-diones: Tumor Cell Apoptosis through tNOX-Modulated NAD(+)/NADH Ratio and SIRT1. Journal of medicinal chemistry 2015, 58(24):9522-9534.

11. Chang CF, Islam A, Liu PF, Zhan JH, Chueh PJ: Capsaicin acts through tNOX (ENOX2) to induce autophagic apoptosis in p53-mutated HSC-3 cells but autophagy in p53-functional SAS oral cancer cells. Am J Cancer Res 2020, 10(10):3230-3247.

12. Lin CY, Islam A, Su CJ, Tikhomirov AS, Shchekotikhin AE, Chuang SM, Chueh PJ, Chen YL: Engagement with tNOX (ENOX2) to Inhibit SIRT1 and Activate p53-Dependent and -Independent Apoptotic Pathways by Novel 4,11-Diaminoanthra[2,3-b]furan-5,10-diones in Hepatocellular Carcinoma Cells. Cancers (Basel) 2019, 11(3).

13. Chen HY, Cheng HL, Lee YH, Yuan TM, Chen SW, Lin YY, Chueh PJ: Tumor-associated NADH oxidase (tNOX)-NAD+-sirtuin 1 axis contributes to oxaliplatin-induced apoptosis of gastric cancer cells. Oncotarget 2017, 8(9):15338-15348.

14. Xu Q, Liu X, Mohseni G, Hao X, Ren Y, Xu Y, Gao H, Wang Q, Wang Y: Mechanism research and treatment progress of NAD pathway related molecules in tumor immune microenvironment. Cancer Cell Int 2022, 22(1):242.

15. Brandes RP, Weissmann N, Schroder K: Nox family NADPH oxidases: Molecular mechanisms of activation. Free Radic Biol Med 2014, 76:208-226.

16. Gong S, Wang S, Shao M: NADPH Oxidase 4: A Potential Therapeutic Target of Malignancy. Front Cell Dev Biol 2022, 10:884412.

17. Wang Y, Sui Y, Niu Y, Liu D, Xu Q, Liu F, Zuo K, Liu M, Sun W, Wang Z et al: PBX1-SIRT1 Positive Feedback Loop Attenuates ROS-Mediated HF-MSC Senescence and Apoptosis. Stem Cell Rev Rep 2023, 19(2):443-454.

18. Wang X, Lu Y, Tuo Z, Zhou H, Zhang Y, Cao Z, Peng L, Yu D, Bi L: Role of SIRT1/AMPK signaling in the proliferation, migration and invasion of renal cell carcinoma cells. Oncol Rep 2021, 45(6).

19. Liu SC, Yang JJ, Shao KN, Chueh PJ: RNA interference targeting tNOX attenuates cell migration via a mechanism that involves membrane association of Rac. Biochem Biophys Res Commun 2008, 365(4):672-677.

20. Lin MH, Lee YH, Cheng HL, Chen HY, Jhuang FH, Chueh PJ: Capsaicin Inhibits Multiple Bladder Cancer Cell Phenotypes by Inhibiting Tumor-Associated NADH Oxidase (tNOX) and Sirtuin1 (SIRT1). Molecules 2016, 21(7).

21. Cheng HL, Lee YH, Yuan TM, Chen SW, Chueh PJ: Update on a tumor-associated NADH oxidase in gastric cancer cell growth. World J Gastroenterol 2016, 22(10):2900-2905.

22. Lee YH, Chen HY, Su LJ, Chueh PJ: Sirtuin 1 (SIRT1) Deacetylase Activity and NAD(+)/NADH Ratio Are Imperative for Capsaicin-Mediated Programmed Cell Death. J Agric Food Chem 2015, 63(33):7361-7370.

23. Islam A, Su AJ, Zeng ZM, Chueh PJ, Lin MH: Capsaicin Targets tNOX (ENOX2) to Inhibit G1 Cyclin/CDK Complex, as Assessed by the Cellular Thermal Shift Assay (CETSA). Cells 2019, 8(10).

24. Su YC, Lin YH, Zeng ZM, Shao KN, Chueh PJ: Chemotherapeutic agents enhance cell migration and epithelial-to-mesenchymal transition through transient up-regulation of tNOX (ENOX2) protein. Biochim Biophys Acta 2012, 1820(11):1744-1752.

25. Zeng ZM, Chuang SM, Chang TC, Hong CW, Chou JC, Yang JJ, Chueh PJ: Phosphorylation of serine-504 of tNOX (ENOX2) modulates cell proliferation and migration in cancer cells. Experimental cell research 2012, 318(14):1759-1766.

26. Chueh PJ, Wu LY, Morre DM, Morre DJ: tNOX is both necessary and sufficient as a cellular target for the anticancer actions of capsaicin and the green tea catechin (-)-epigallocatechin-3-gallate. Biofactors 2004, 20(4):235-249.

27. Ran D, Zhou D, Liu G, Ma Y, Ali W, Yu R, Wang Q, Zhao H, Zhu J, Zou H et al: Reactive Oxygen Species Control Osteoblast Apoptosis through SIRT1/PGC-1alpha/P53(Lys382) Signaling, Mediating the Onset of Cd-Induced Osteoporosis. J Agric Food Chem 2023.

28. Zhang Z, Chen X, Liu S: Role of Sirtuin-1 in Neonatal Hypoxic-Ischemic Encephalopathy and Its Underlying Mechanism. Med Sci Monit 2020, 26:e924544.

29. McReynolds MR, Chellappa K, Baur JA: Age-related NAD(+) decline. Exp Gerontol 2020, 134:110888.

30. Xie N, Zhang L, Gao W, Huang C, Huber PE, Zhou X, Li C, Shen G, Zou B: NAD(+) metabolism: pathophysiologic mechanisms and therapeutic potential. Signal Transduct Target Ther 2020, 5(1):227.

31. Zapata-Perez R, Wanders RJA, van Karnebeek CDM, Houtkooper RH: NAD(+) homeostasis in human health and disease. EMBO Mol Med 2021, 13(7):e13943.

32. Wang B, Ma Y, Kong X, Ding X, Gu H, Chu T, Ying W: NAD(+) administration decreases doxorubicin-induced liver damage of mice by enhancing antioxidation capacity and decreasing DNA damage. Chem Biol Interact 2014, 212:65-71.

33. Hong G, Zheng D, Zhang L, Ni R, Wang G, Fan GC, Lu Z, Peng T: Administration of nicotinamide riboside prevents oxidative stress and organ injury in sepsis. Free Radic Biol Med 2018, 123:125-137.

34. Nadysev GY, Tikhomirov AS, Lin MH, Yang YT, Dezhenkova LG, Chen HY, Kaluzhny DN, Schols D, Shtil AA, Shchekotikhin AE et al: Aminomethylation of heliomycin: Preparation and anticancer characterization of the first series of semi-synthetic derivatives. European journal of medicinal chemistry 2018, 143:1553-1562.